# Pruning's Effect on Generalization Through the Lens of Training and Regularization

**Tian Jin**[1][†] **Michael Carbin**[1] **Daniel M. Roy**[2] **Jonathan Frankle**[3] **Gintare Karolina Dziugaite**[4]

[1]MIT  [2]University of Toronto, Vector Institute  [3]MosaicML  [4]Google Research, Brain Team

## Abstract

Practitioners frequently observe that pruning improves model generalization. A long-standing hypothesis based on bias-variance trade-off attributes this generalization improvement to model size reduction. However, recent studies on over-parameterization characterize a new model size regime, in which larger models achieve better generalization. Pruning models in this over-parameterized regime leads to a contradiction – while theory predicts that reducing model size harms generalization, pruning to a range of sparsities nonetheless improves it. Motivated by this contradiction, we re-examine pruning's effect on generalization empirically.

We show that size reduction cannot fully account for the generalization-improving effect of standard pruning algorithms. Instead, we find that pruning leads to better training at specific sparsities, improving the training loss over the dense model. We find that pruning also leads to additional regularization at other sparsities, reducing the accuracy degradation due to noisy examples over the dense model. Pruning extends model training time and reduces model size. These two factors improve training and add regularization respectively. We empirically demonstrate that both factors are essential to fully explaining pruning's impact on generalization.

## 1 Introduction

Neural network pruning techniques remove unnecessary weights to reduce the memory and computational requirements of a model. Practitioners can remove a large fraction (often 80-90%) of weights without harming *generalization*, measured by test error [31, 19, 13]. While recent pruning research [19, 13, 53, 7, 30, 44, 11, 67, 4, 34, 65, 54, 35, 37, 57, 55, 9, 68, 60] focuses on reducing model footprint, improving generalization has been a core design objective for earlier work on pruning [31, 22]; recent pruning literature also frequently notes that it improves generalization [19, 13].

How does pruning affect generalization? An enduring hypothesis claims that pruning may benefit generalization by reducing *model size*, defined as the number of weights in a model.[1] We refer to this as the *size-reduction hypothesis*. However, algorithms for pruning neural networks and our understanding of the impact of model size on generalization have changed.

First, pruning algorithms have grown increasingly complex. Learning rate rewinding [53], a state-of-the-art algorithm prunes weights iteratively. An iteration consists of *weight removal*, where the

---

[†]Correspondence to: `tianjin@csail.mit.edu`, `mcarbin@csail.mit.edu`, `gkdz@google.com` .

[1]This hypothesis traces back to seminal work in pruning, such as Optimal Brain Surgeon [22]: "without such weight elimination, overfitting problems and thus poor generalization will result." And again in pioneering work [19] applying pruning to deep neural network models: "we believe this accuracy improvement is due to pruning finding the right capacity of the network and hence reducing overfitting." Because the latter hypothesis leaves the definition of network capacity unspecified, we examine an instantiation of it, measuring network capacity with the number of weights in a neural network model.

36th Conference on Neural Information Processing Systems (NeurIPS 2022).

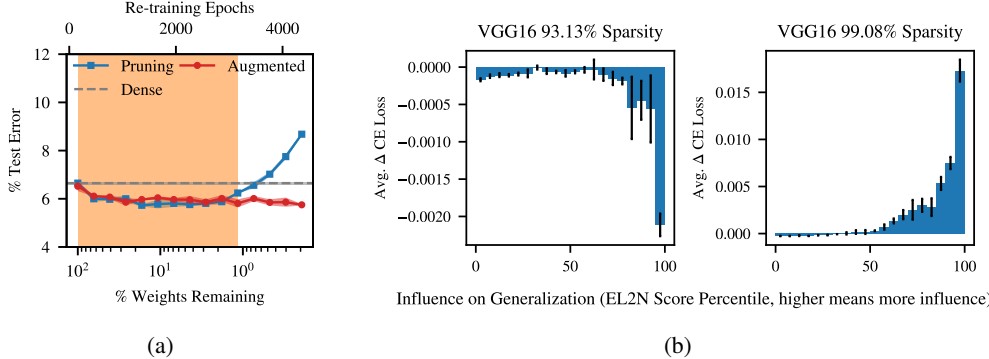

(a)                                                                    (b)

Figure 1: (a) Retraining the VGG-16 model on CIFAR-10 exactly as done in pruning, but without removing any weights produces models with similar generalization as pruning to the shaded range of weights remaining. We show the x-axis for pruning/augmented training below/above the plot respectively. (b) We show the loss difference between the sparse/dense model produced by pruning/standard training respectively for training examples grouped by EL2N score percentiles. Negative value indicates that the sparse model attains better loss. Left: We find that pruning to a relatively low sparsity may improve generalization by decreasing training loss, with a particular decrease on influential examples. Right: We find that pruning to a relatively high sparsity may also improve generalization by increasing training loss, also with a particular increase on influential examples.

algorithm removes a subset of remaining weights, followed by *retraining*, where the algorithm continues to train the model, replaying the original learning rate schedule. Across iterations, this retraining scheme effectively adopts a cyclic learning rate schedule [58], which may benefit generalization.

Second, emerging empirical and theoretical findings challenge our understanding of generalization for deep neural network models with an ever-increasing number of weights [48, 12, 46, 69]. In particular, the size-reduction hypothesis builds on the classical generalization theory on *bias-variance trade-off*, which predicts that reducing model size improves the generalization of an overfitted model [23]. However, recent work [5, 47] reveals that beyond this classical bias-variance trade-off regime lies an over-parameterized regime where models are large enough to achieve near-zero training error. In this regime, bigger models often achieve better generalization [5, 47, 29, 27, 59]. Practitioners routinely apply modern pruning algorithms to models in this over-parameterized regime. While our renewed understanding of generalization predicts that reducing the size of an over-parameterized model should harm generalization, pruning to a range of sparsities nevertheless improves it. The size-reduction hypothesis is therefore inconsistent with our empirical observation about pruning.

**A new direction.** We first confirm that the size-reduction hypothesis does not fully explain pruning's effect on generalization. Figure 1a compares the generalization of a family of sparse models of different sizes generated by a state-of-the-art pruning algorithm [53], and the generalization of a family of models generated by the same algorithm, except modified to no longer remove weights. This algorithm, which we refer to as the *augmented training algorithm*, differs from standard training in two ways: (1) the number of training epochs is larger; (2) the learning rate schedule is cyclic.

The results show that for any sparse VGG-16 model generated by the pruning algorithm with more than 1% of its original weights, its generalization is indistinguishable from that of the model generated by the augmented training algorithm that does not remove weights but trains the model for the same number of epochs and with the same learning rate schedule as the pruning algorithm.[2] For a range of sparsities, pruning's effect on generalization remains unchanged in the absence of weight removal, illustrating that the size-reduction hypothesis cannot fully explain pruning's effect on generalization.

**Our approach.** We instead develop an explanation for pruning's impact on generalization through an analysis of its effect on each training example's loss – the difference in the example's training loss between the sparse model produced by pruning and the dense model produced by standard training.

---

[2] In Appendix C we show this phenomenon for 4 additional benchmarks, and other pruning algorithms.

We then interpret pruning's effect on each example in relation to the example's influence on generalization. In particular, an example's influence on generalization is the absolute change to generalization due to leaving this example out, which Paul et al. [51] propose to approximate with the L2 distance between the predicted probabilities and one-hot labels early in training, called the EL2N (**Error L2 Norm**) score. Paul et al. [51] show that an example with a high EL2N score may cause a large weight update, thereby substantially changing the predictions and loss of other examples. We thus refer to high EL2N examples as *influential examples*. Our analysis shows the following results.

**Better training.**   We prune a VGG-16 model on the CIFAR-10 dataset to the *optimal sparsity*, the sparsity with the best validation error. At this sparsity, we find that the pruned model achieves better training loss than the dense model. In the left plot in Figure 1b, we present the change in training loss due to pruning for the training examples grouped by their EL2N score percentiles. We observe that influential examples show the greatest training loss improvement. Pruning may therefore lead to *better training*, improving the training loss of the sparse model over the dense model, particularly on examples with a large influence on generalization.

**Additional regularization.**   We prune the same VGG-16 model on CIFAR-10 dataset with the pruning algorithm configured to produce the sparsest model that still improves generalization over the dense model. At this sparsity, we find that the pruned model achieves worse training loss than the dense model. In the right plot in Figure 1b, we present the change in training loss due to pruning for the training examples grouped by their EL2N score percentiles. The results show that pruning may also improve generalization while increasing the training loss of the sparse model over the dense model, particularly on examples with a large influence on generalization.

While it is perhaps counter-intuitive that increasing training loss, particularly on the most influential examples may improve generalization, examining these pruning-affected examples leads us to hypothesize that it is due to an observation that Paul et al. [51] made: in image classification datasets, a small fraction of influential training examples have ambiguous or erroneous labels, training on which impairs generalization. Therefore, it is instead better for the model to not fit them.

We examine this hypothesis by introducing random label noise into the training dataset. We find that on such datasets, pruning may improve generalization by increasing the training loss on the noisy examples, while still preserving the training loss of examples from the original training dataset. The result is an overall generalization improvement for the sparse model. Notably, reducing model width has a similar effect on training loss and generalization, suggesting a reduced number of weights in the model as the underlying cause for generalization improvement in the presence of noisy examples. We refer to the effect of removing weights as *additional regularization*, which reduces the accuracy degradation due to noisy examples over the original dense model.

**Implications.**   It has been a long-standing hypothesis that pruning improves generalization by reducing the number of weights (via the Minimum Description Length principle [18] or Vapnik–Chervonenkis theory [63]). Our empirical observations suggest that the theory and practice of pruning should instead focus on the two effects of modern pruning algorithms: better training and additional regularization, both of which are indispensable to explaining pruning's benefits to generalization fully. Our results also suggest that quantifying pruning's heterogenous effect across training examples is key to understanding pruning's influence on generalization.

**Contributions.**   We present the following contributions:

1. We find that pruning may lead to *better training*, improving the training loss, as well as the generalization over the dense model at optimal sparsity.

2. We find that pruning may also lead to *additional regularization*, reducing the accuracy degradation due to noisy examples, and improving the generalization over the dense model at optimal sparsity.

3. We demonstrate that our deconstruction of pruning's effect into training and regularization cannot be further simplified — alternative explanations such as extended training time and reduced model size only partially explain the effects of pruning on generalization.

## 2 Preliminaries

We study *Iterative Magnitude Pruning (IMP) with learning-rate rewinding* [53]. This algorithm consists of four steps: (1) Train a neural network to completion. (2) Remove a specified fraction of the smallest-magnitude remaining weights. (3) Reset the learning rate to its value in an earlier epoch $t$ and (4) Repeat steps 1-3 iteratively until the model reaches the desired overall sparsity.

**Terminology.** *Dense models* refer to models without any removed weights. *Sparse models* refer to models with removed weights as a result of pruning. The *optimally sparse model* is the model pruned to the sparsity at which it achieves the best validation error. The *optimal sparsity* is the sparsity of the optimally sparse model. *Generalization* refers to the classification error on the test set.

**Experimental methodology.** We use standard architectures: LeNet [32], VGG-16 [56], ResNet-20, ResNet-32 and ResNet-50 [24], and train on benchmarks (MNIST, CIFAR-10, CIFAR-100, ImageNet) using standard hyperparameter settings and standard cross-entropy loss function [13, 14, 66]. Following Frankle and Carbin [13], Frankle et al. [14], we set the $t$ in IMP to $t = 0$ for MNIST-LeNet benchmark and $t = 10$ for the others. Appendix B shows further details.

We report all results by running the same experiment 3 times with distinct random seeds. We conclude that any real-valued results of two experiments differ if and only if the mean difference between three independent runs of respective experiments is at least one standard deviation away from zero, otherwise we say that the results *match*. We use PyTorch [50] on TPUs with OpenLTH library [13].

**Limitations.** Our work is empirical in nature. Though we validate each of our claims with extensive experimental results on benchmarks with different architectures and datasets, we recognize that our claims may still not generalize due to the empirical nature of our study.

## 3 Generalization Improvement from Better Training

Not all training examples have the same effect on training and generalization [62, 51, 8, 21, 25]. In this section, we measure pruning's effect on each training example by examining the difference in its training loss between the dense model produced by standard training and the sparse model produced by pruning. Our results show that pruning most significantly affects training examples that are most influential to model generalization.

**Method.** Paul et al. [51] propose to estimate an example's influence on model generalization with the EL2N (**E**rror **L2 N**orm) score – the L2 distance between the predicted probability and the one-hot label of the example early in training. Details for computing the EL2N score is available in Appendix B.3. They show that an example with a high EL2N score may cause a large weight update, thereby substantially changing the predictions and loss of other examples. Paul et al. [51] rely on this relationship between EL2N score and generalization to remove examples with little influence on generalization from the dataset to accelerate training without affecting model generalization. We similarly use the EL2N score to assess the impact of pruning on generalization.

To measure the difference in training loss on examples due to pruning, we partition the training set into $M$ subgroups, each with a different range of EL2N score percentiles. We then compute the average training loss difference on examples in each subgroup. We compute this difference by subtracting the training loss of the dense model produced by standard training from that of the sparse model produced by pruning. A negative value indicates that the sparse model has better training loss than the dense model. We pick $M = 20$ because it is the largest value of $M$ that enables us to clearly present the per-subgroup training loss difference.

We measure training loss at two sparsities of interest: the sparsity that achieves the best generalization, and the highest sparsity that still attains better generalization than the dense model. In Appendix L, we present pruning's effect on subgroup training loss at a wider range of sparsities.

**Results.** Figure 2a presents pruning's effect on subgroup training loss. Our results show that at the sparsity that achieves the best generalization, for all subgroups, the average training loss of the pruned model either matches or improves over the dense model, indicating better training.

Figure 2a also shows that pruning's effect on training example subgroups is nonuniform. Pruning most significantly affects the most influential examples – the ones with the highest range of EL2N scores. Specifically, the average magnitude of pruning's effect on examples with the top 20% EL2N

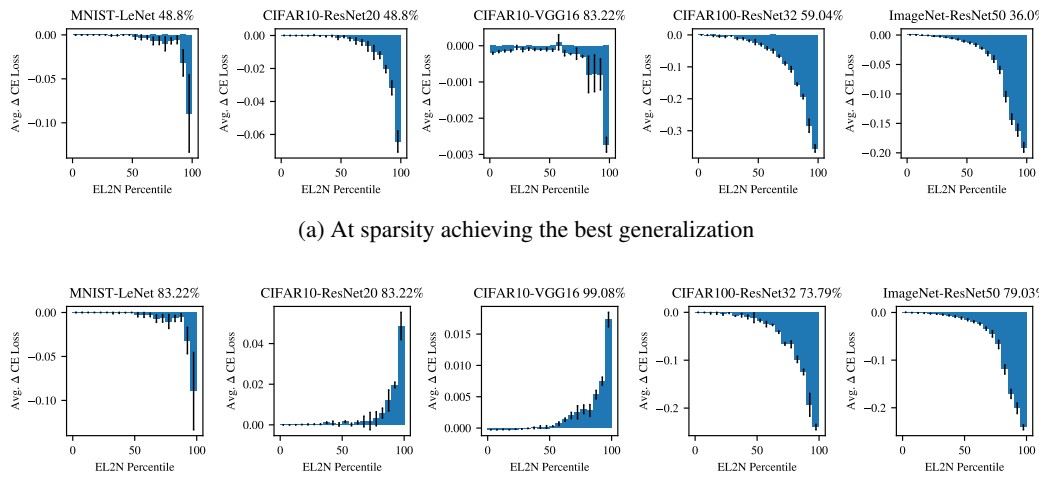

(a) At sparsity achieving the best generalization

(b) At highest sparsity with generalization exceeding that of the dense counterpart

Figure 2: Difference between the average training loss of the dense model produced by standard training and the sparse model produced by pruning on example subgroups with distinct EL2N percentile range. Negative values indicate that the sparse model has better loss than the dense model.

scores is 3.5-71x that on examples with the bottom 80% EL2N scores. Pruning to the said sparsity thus improves generalization while improving the training loss of example subgroups nonuniformly, with an emphasis on influential examples. In Appendix D, we validate the effect of pruning-improved examples on generalization: excluding 20% of pruning-improved examples hurts generalization more than excluding a random subset of the same size.

**Conclusion.** On standard datasets, pruning to the sparsity that achieves the best generalization leads to better training – the training loss of most example subgroups improves over the dense model. The subgroup with the most influence on generalization sees the largest training loss improvement.

## 4    Generalization Improvement from Additional Regularization

Pruning to the highest sparsity that improves generalization (Figure 2b) displays a similar effect as pruning to the sparsity that attains the best generalization (Figure 2a) – generalization improves with an overall improved training loss over the dense model, except for CIFAR-10, where pruning improves generalization while worsening training loss on most subgroups of examples. Specifically, for ResNet20 on CIFAR-10 at 83.22% sparsity (Figure 2b, second from left) and VGG-16 on CIFAR-10 at 99.08% sparsity (Figure 2b, third from left), the training loss of the sparse model is worse than the dense model. While pruning always increases training loss when it removes a large enough fraction of weights, the accompanied generalization improvement does not always occur in general.

The second and third plots of Figure 2b show that the loss increase is especially pronounced on particular subgroups of influential examples. Examining these pruning-affected training examples in Appendix K.1 leads us to hypothesize that the generalization improvement is due to an observation Paul et al. [51] made: although influential examples are predominately beneficial for generalization, there exist a small fraction of noisy examples, such as ones with ambiguous or erroneous labels, that exert a significant influence on generalization, albeit in a harmful way.[3]

To precisely characterize pruning's effect on noisy examples, we inject random label noise into the training dataset. Doing so enables a comparison of pruning's effect on noisy data and original data.

**Method.** We inject $p\%$ random label noise by selecting $p\%$ examples uniformly at random and changing the label of each example to one of the other labels in the dataset sampled uniformly at random. We refer to the data that is not affected as original data. We sample and fix the random labels before each independent run of an experiment.

---

[3]We present leave-subgroup-out retraining experiments for these examples in Appendix K.

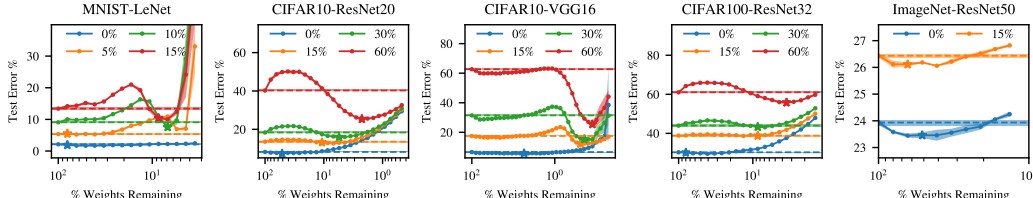

Figure 3: Pruning improves generalization over the dense model in the presence of random label noise. We show the generalization of the sparse models as a function of the % weights remaining. Horizontal dashed lines indicate the baseline generalization of the dense models. We mark the optimal sparsity with a star. Legends show the fraction of the training examples with random label noise.

We report the following results for models trained on datasets with and without random label noise: (1) pruning's impact on generalization versus sparsity (Figure 3); (2) pruning's impact on training loss for the original data and random label data versus sparsity (Figure 4);[4] (3) pruning's impact on training loss for subgroups of examples, each with a different range of EL2N scores (Figure 5).

**Generalization results.** Figure 3 shows that, in the presence of random label noise, the generalization of the optimally sparse models is better than that of the dense models on 11 out of 13 benchmarks and matches that of the dense models on the rest of the benchmarks. Our results provide evidence that pruning to the optimal sparsity improves generalization in the presence of random label noise.

The extent of pruning's generalization improvement at the optimal sparsity grows as the portion of label noise increases. For example, for VGG-16 on CIFAR-10 benchmark, pruning to the optimal sparsity improves generalization by 5.2%, 16.1% and 34.6% on 15%, 30% and 60% label noise.

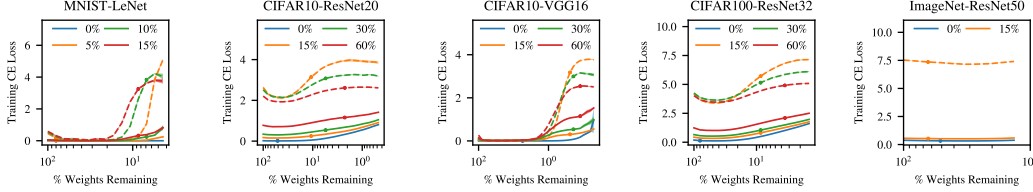

Figure 4: Pruning to the optimal sparsity increases training loss over the dense model in the presence of random label noise. Noisy examples see a particular large loss increase. We show the training loss on the original and noisy data with dashed and solid lines respectively, as functions of % weights remaining. We mark the optimal sparsity with a star. Legends show the random label noise level.

**Original versus noisy data.** Figure 4 shows that pruning initially reduces the average training loss on noisy examples: for instance, the training loss on noisy examples is slightly lower for the sparse ResNet20 models on CIFAR-10 with many weights remaining (i.e., more than 10% ) than for the dense model. Since noisy examples are influential examples [51], these results indicate better training, which improves model training loss with a particular emphasis on influential examples.

As the fraction of weights remaining drops further, the loss of the sparse model on noisy data grows. For instance, the training loss on noisy examples increases for the sparse ResNet20 models on CIFAR-10 with few weights remaining (i.e., less than 10%). Moreover, the gap between the average training loss of the sparse model on original versus noisy examples widens beyond the dense model, indicating that pruning leads to additional regularization. In particular, for a range of sparsities where generalization improves over the dense model, the sparse model fits original examples significantly better than those with randomized labels. Taking the dense VGG-16 model trained on the CIFAR-10 dataset with 15% – 60% random label noise as an example, we observe that this model incurs 0.03 – 0.13 higher training loss on the partition with injected random label noise than on the partition without. Pruning to the optimal sparsity widens this training loss gap to 1.38 – 2.97. Furthermore, the gap between the average training loss of the sparse models on original versus noisy examples eventually diminishes as the sparsity increases even further.

---

[4]Numerical results available in Table 11 in Appendix M

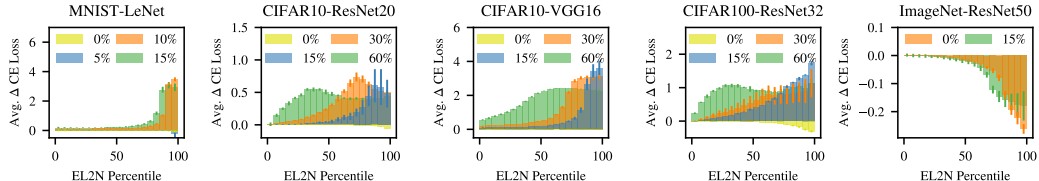

Figure 5: In the presence of random label noise, pruning to the optimal sparsity often increases training loss (except for ResNet50) over the dense model, especially to highly influential examples. For example subgroups with distinct EL2N percentile ranges, we show the difference in average training loss between dense and optimally sparse models. Legend indicates label noise level. We do not show LeNet with 5% noise because the dense model achieves the best generalization. For ResNet20 and ResNet32 with 60% noise, the training loss for high EL2N examples are already very high for dense models to begin with, leaving little room for their loss to increase further by pruning.

**Difference in loss by subgroups.** Figure 5 shows the difference between the training loss of the dense and optimally sparse model on subgroups of examples with different EL2N scores, but this time in the presence of random label noise. The results show that pruning to the optimal sparsity often increases the loss of influential examples, contrary to what Figure 2 shows when training on original data only where pruning decreases the training loss of influential examples.

As random label examples are influential and harm generalization [51], increasing the training loss on influential examples may therefore improve generalization. In Appendix E, we validate that excluding examples misclassified by the optimally sparse models improves generalization similar to pruning.

**Conclusion.** We study pruning's effects on training and regularization by connecting them to noisy examples. By introducing random label noise, we show the two effects of pruning more clearly. At relatively low sparsities, we observe better training because the training loss of sparse models is lower than the dense model, especially on influential examples highly influential to generalization. As sparsity increases, regularization effects dominate, because the sparse models see a larger gap between training loss on the noisy versus original data than the dense model. We note that across all sparsities, pruning has a highly nonuniform effect across example subgroups with different influences on generalization (EL2N scores) on both standard datasets and datasets with random label noise.

Crucially, the effect of pruning at the optimal sparsity – be it better training or additional regularization – depends on the noise levels in the dataset. On datasets with few noisy examples, Figure 2 shows that at the optimal sparsity, better training is the key to good generalization, because training loss reduces. On datasets with injected random label noise, Figure 5 shows that the optimal sparsity is the one at which pruning-induced regularization effects increase noisy data loss.

## 5   Isolating the Effects of Extending Training Time and Reducing Model Size

We identified two effects of pruning on generalization: better training and additional regularization. They intersect with two components of the pruning algorithm: extended training time and model size reduction, respectively. In this section, we study these two effects in isolation by extending dense model training time and reducing the dense model size by scaling down its width.[5]. We confirm that both effects of pruning are necessary to fully explain pruning's impact on generalization. Thus our deconstruction of pruning's effect in terms of training and regularization cannot be further simplified.

### 5.1   Extended Training Time

What happens to generalization if the weight removal part is removed from pruning, and only the retraining part remains? To answer, we train dense models exactly as done in the pruning algorithm, for the same number of gradient steps and learning rate schedule, but without removing weights.

---

[5]To isolate other effects of pruning, we also evaluate the contribution of design choices of pruning algorithms, such as the heuristic selecting which weights to prune, to generalization via ablation studies in Appendix J

**Method.** We study the effect of training dense models for extra gradient steps on generalization, without removing any weights. When training the dense models, we use the same learning rate schedule as pruning, replaying the original learning rate schedule designed for the dense model multiple times, effectively adopting a cyclic learning rate schedule. We refer to this training algorithm as *extended dense training*. We report and compare the generalization of models that pruning and extended dense training algorithm produce.

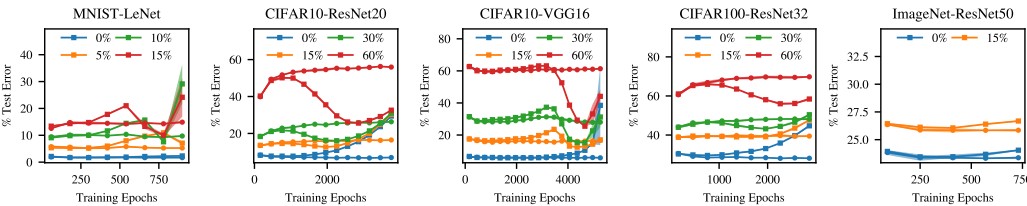

Figure 6: Extended training time does not explain the full extent of pruning's benefits to generalization. We show the test errors of the dense and sparse models with circle and square markers respectively as a function of the training epochs. Legends show random label noise level.

**Results.** Figure 6 shows test errors of models that extended dense training and pruning produce as a function of training epochs.[6] The two algorithms produce models with similar generalization on standard datasets without random label noise, where pruning improves training at optimal sparsity, as shown in Section 3. Indeed, in Appendix L, we show that longer training time similarly leads to better training. However, on benchmarks with random label noise, extended dense training underperforms pruning on 9 out of the 13 benchmarks. On LeNet, ResNet20, VGG-16, ResNet32 and ResNet50 benchmarks with noise, extended dense training achieves test errors that are worse than pruning by 0 (matching) to 2.6%, 0.4 to 14.5%, 3.6 to 34.2%, 0 (matching) to 4.8% and -0.3%, respectively.

**Conclusion.** On standard datasets without random label noise, the extended dense training algorithm produces models with generalization that matches or exceeds the optimally sparse model. However, with random label noise, the extended dense training algorithm can no longer produce models that match the generalization of the optimally sparse model.

## 5.2 Size Reduction

What happens to generalization if the retraining part is removed from pruning, and only the weight removal part remains? We compare pruning with *width down-scaling* – applying standard training to dense models with reduced widths. The latter removes weights without the need for retraining.

At optimal sparsity, we show that size reduction is not necessary to replicate pruning's generalization improvements with dense models in Figure 1a from Section 1; in this section, we show that size reduction is also not sufficient to replicate pruning's generalization improvement with dense models.

**Method.** We reduce the model size by down-scaling the width of a dense model: we train a sequence of dense models using the standard number of training epochs, where the next model in the sequence has 80% of the width of the model preceding it. This width scaling ratio mimics our sparsity schedule, where the pruned model always has 80% of the remaining weights in the previous iteration. We then compare the generalization of models that pruning and width down-scaling produce.

**Results.** Figure 7 presents the test errors of models that pruning and width down-scaling produce, using square and circle markers respectively. Width down-scaling under-performs pruning on 4 out of 5 benchmarks on datasets without random label noise. When training with random label noise, width down-scaling has notable regularization effects and significantly improves generalization, similar to pruning. In Appendix I, we show that the regularization effects of both algorithms manifest similarly across example subgroups, suggesting that pruning's regularization effect is a consequence of model size reduction in general. However, despite this improvement, a gap remains between the minimum test error of models that pruning and width down-scaling produce achieve: pruning does better on LeNet, ResNet20, VGG-16, ResNet32 and ResNet50 benchmarks by -4 (pruning does worse) to 0.4%, 0% (matching), 1.5 to 1.8%, 0 (matching) to 2.1%, and 0.36% in test error, respectively.

---

[6]Numerical results available in Table 5 in Appendix M.

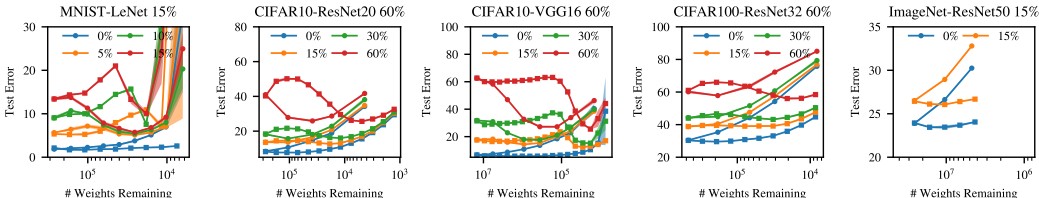

Figure 7: Size reduction does not explain the full extent of pruning's benefits to generalization. We show the test errors of the dense/sparse models with round/square markers respectively as a function of the weights remaining. Legends show label noise level. Appendix F includes detailed views.

**Conclusion.** On standard datasets with random label noise, width down-scaling produces models with similar generalization (no worse by more than 2.1%) as the optimally sparse models. However, without random label noise, width down-scaling under-performs pruning.

## 5.3 Conclusion

By removing the weight removal part from pruning in Section 5.1, we show the importance of pruning-induced additional regularization for achieving good generalization in the presence of label noise more clearly than Section 4. Similarly, by removing the retraining part from pruning in Section 5.2, we show the importance of pruning-induced better training for achieving good generalization using the original dataset without random label noise more clearly than Section 3.

We confirm that both effects of pruning are required to fully account for its impact on generalization. Our deconstruction of pruning's effect in terms of training and regularization is therefore minimal.

## 6 Related Work

Pruning has an extended history: early work starting from the 1980s found that pruning enhances model interpretability and generalization [45, 31, 22]. More recently, the advent of deep neural network models motivated the adoption of pruning to reduce the storage and computational demand of deep models [19, 20, 30, 40, 7, 44, 11, 67, 4, 34, 65, 54, 35, 37, 57, 55, 9, 68, 60]. In this section, we describe several branches of pruning research pertinent to our work.

**Pruning algorithm design.** Recent pruning research often focuses on improving pruning algorithm design. An extended line of research studied the heuristics that determine which weights to remove [44, 11, 33] – the simplest heuristic is to remove weights with the smallest magnitude. For example, Molchanov et al. [43], Louizos et al. [42] proposed to learn which weights to prune as part of the optimization process. Another line of research concerned the structure of the weights to remove to achieve computational speedup without significantly hurting model generalization. For example, Li et al. [36], Liu et al. [41] proposed to prune weights in groups or to prune neurons, convolutional filters and channels. Our work does not produce a new pruning algorithm design. Rather, we study the existing pruning algorithms to understand their effect on model generalization.

**Pruning's effect on generalization.** There are many reported instances in which pruning benefits model generalization [31, 19, 13]. This effect has sparked interest in examining pruning as a technique to gain additional generalization benefits beyond common regularization techniques. *Weight decay* is a typical regularization technique that encourages model weights to have a small $L_2$ norm as part of the training objective. Giles and Omlin [16] demonstrated that pruning can improve the generalization of recurrent neural network models better than weight decay. Thimm and Fiesler [61], Augasta and Kathirvalavakumar [1] compared the effect on the generalization of several contemporary pruning techniques. Bartoldson et al. [3] discovered a positive correlation between *pruning instability*, defined as the drop in test accuracy immediately following weights removal, and model generalization. The authors found that higher pruning-induced instability leads to increased flatness of minima, which in turn improves generalization. Our work differs from prior work as we study pruning's effects on generalization through a novel perspective, examining pruning's impact on training examples.

**Pruning's effect beyond generalization.** Our work contributes to a growing line of work [25, 38] investigating pruning's effect on examples beyond test error. Hooker et al. [25] discovered that model compression can have a disproportionately large impact on predicting the under-represented long-tail of the data distribution. Liebenwein et al. [38] studied the performance of pruning using metrics such as out-of-distribution generalization and resilience to noise, and found that pruning may not preserve these alternative performance metrics even when it preserves test accuracy. To the best of our knowledge, our work is the first to examine pruning's impact on examples in the training set.

## 7   Closing Discussion

**Importance of training improvement.** Our work sheds light on an overlooked and underestimated effect of pruning that boosts generalization – training improvement. Seminal work on pruning [22, 19] focuses on its regularization effect and attributes pruning's beneficial influence on generalization to regularization that comes with model size reduction. Instead, our work discovers that, on standard image classification benchmarks, the state-of-the-art pruning algorithm attains the optimal generalization consistently by improving model training.

**How does model size reduction impact generalization?** The relationship between *model complexity* and generalization has long been a subject of immense interest [52, 23, 5, 47]. Informally, model complexity refers to the model's ability to fit a wide variety of functions [17]. For example, model size [5] and the norm of model weights [49] both provide meaningful measures of model complexity for the study of generalization. Classical theories (e.g., on VC dimension [64] and Rademacher complexity [2]) show that reducing model complexity decreases the likelihood for the learning algorithm to produce trained models with a large gap between training and test loss, and may therefore improve generalization. We instead focus on learning dynamics of training example subgroups and show that reducing model size improves generalization by mitigating the accuracy degradation due to noisy examples. Therefore, our work contributes a complementary perspective to the understanding of the relationship between model complexity and generalization.

**Societal impact.** Our work examines pruning's effect on generalization. While we measure generalization using test error, Hooker et al. [25] show that test error may be an imperfect characterization of model prediction quality due to a lack of consideration of fairness and equity. Concretely, our work does not address the potential disproportionate impact of model pruning on label categories. Our work must therefore be interpreted within the context of the community's current and future understanding of pruning's potential contribution to systemic bias, especially against minority groups that may be underrepresented in existing training datasets.

**Implication for fairness.** A growing line of work concerns the adverse impact of pruning on model fairness [25, 26, 6, 39] – they show that pruning disproportionately worsens the prediction accuracy on a small subgroup of examples. However, our results show that their work does not completely characterize pruning's effects pertinent to fairness: (1) We show that only at relatively high sparsities can pruning harm the prediction accuracy of any example subgroup, where pruning's regularization effects dominate. (2) We show that size reduction in general, rather than pruning in particular, underpins the disproportionately worse accuracy of the pruned model on certain subgroups of examples. Our results call for further assessment and discussion of pruning's risk to model fairness.

**Conclusion.** We show that the long-standing size-reduction hypothesis attributing pruning's beneficial effect on generalization to its reduction of model size does not fully explain pruning's impact on generalization. Inspired by studies [62, 51] that show nonuniform effects of training examples on generalization, we develop an analysis to study pruning's impact on generalization by interpreting its effect on an example's training loss in relation to the example's influence on generalization.

With this novel analysis, we find that, at the optimal sparsity, pruning leads to either better training or additional regularization, which improves training loss over the dense model, and reduces the accuracy degradation due to noisy examples over the dense model, respectively. Both effects contribute to improving model generalization.

Our novel analysis adds to our empirical toolkit for studying the effect of a learning algorithm on generalization through its effect on the loss of training examples. Using our novel analysis, we derive findings that advance our empirical understanding of pruning as a learning algorithm.

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
