| Model | Dataset | Epochs | Batch | Opt. | LR | LR Drop | Weight Decay | Initialization | Rewind Epoch |
|-------|---------|--------|-------|------|-----|---------|--------------|----------------|--------------|
| LeNet | MNIST | 60 | 128 | Adam($\beta_1 = 0.9$ $\beta_2 = 0.999$, $\epsilon = 1e - 8$) | 1.2e-3 | - | - | Kaiming Normal | 0 |
| ResNet20 | CIFAR-10 | 160 | 128 | SGD($\mu = 0.9$) | 0.1 | 10x at epoch 80, 120 | 1e-4 | Kaiming Normal | 10 |
| VGG-16 | CIFAR-10 | 160 | 128 | SGD($\mu = 0.9$) | 0.1 | 10x at epoch 80, 120 | 1e-4 | Kaiming Normal | 10 |
| ResNet32 | CIFAR-100 | 160 | 128 | SGD($\mu = 0.9$) | 0.1 | 10x at epoch 80, 120 | 1e-4 | Kaiming Normal | 10 |
| ResNet50 | ImageNet | 90 | 768 | SGD($\mu = 0.9$) | 0.3 | 10x at epoch 30, 60, 80 | 1e-4 | Kaiming Normal | 10 |

Table 1: We use standard hyperparameters following the precedent of Frankle and Carbin [13], Frankle et al. [14], Wang et al. [66]. $\mu$ in SGD configuration parameter denotes momentum.

# A    Acknowledgement.

We thank Zack Ankner, Xin Dong, Zhun Liu, Jesse Michel, Alex Renda, Cambridge Yang, and Charles Yuan for their helpful discussion and feedback to this project. This work was supported in part by Apple, a Facebook Research Award, the MIT-IBM Watson AI-LAB, Google's Tensorflow Research Cloud, and the Office of Naval Research (ONR N00014-17-1-2699). Daniel M. Roy is supported in part by an NSERC Discovery Grant and Canada CIFAR AI Chair funding through the Vector Institute. Part of this work was done while Gintare Karolina Dziugaite and Daniel M. Roy were visiting the Simons Institute for the Theory of Computing.

# B    Additional Experimental Details

We complement the description of experimental methods in Section 2 with additional details.

## B.1    Models and Datasets

We study pruning's effect on generalization using LeNet [32], VGG-16 [56], ResNet-20, ResNet-32 and ResNet-50 [24]. We use the MNIST dataset, consisting of 60,000 images of handwritten digits whose labels correspond to 10 integers between 0 and 9. We also use the CIFAR-10 and -100 [28] datasets, which consists of 60,000 images in 10 and 100 classes. For both datasets, we draw 2,000 of the original training images randomly as validation set; we continue to use the remaining 48,000 of the training images as training set. We use all 10,000 original test images as our test set. We use the ImageNet dataset [10] as well, which contains 1,281,167 images in 1,000 classes. We again randomly draw 50,000 images as validation set and use the remaining 1,231,167 training images as our training set. We use all 50,000 original test images as our test set.

## B.2    Training Hyperparameters

Our training hyper-parameter configuration follows the precedent of Frankle and Carbin [13], Frankle et al. [14], Wang et al. [66]: details are available in Table 1.

## B.3    EL2N Score Calculation

We follow the method that Paul et al. [51] describes to compute the EL2N scores of examples in the training dataset. For a model architecture $f$ and dataset $S$, we first train $N$ of a model for $K\%$ of the total training time to obtain partially trained weights $\theta_n, n = 1, \cdots, N$ for each model. Subsequently, we compute the EL2N score for each image-label pair $(x, y)$, as $\frac{1}{N} \sum_n \|p_{\theta_n}(x) - y\|_2$, where $y$ is one-hot label, and $p_{\theta_n}(x)$ is the softmax output of the model. Following the precedent of [51], we take $N = 10$ and $K = 10$. There is no precedent for choosing the appropriate value for $N, K$ for the ImageNet benchmark, and we find setting $N = 22$, which corresponds to measuring EL2N scores at the 20th epoch of training, and $K = 10$ to be a reasonable choice.

# C    Size-Reduction Does Not Explain Pruning's Benefits to Generalization

In Section 1, we develop an augmented version of the pruning algorithm, by modifying the pruning algorithm to no longer remove weights. This augmented training algorithm corresponds to extended

dense training with a cyclic learning rate schedule (EDT) we examine in Section 5.1 – it trains the model for the same number of epochs and with the same cyclic learning rate schedule as pruning.

In Section 1, we highlight the similarity between the generalization of models that this EDT algorithm and pruning produce. This similarity shows that the size-reduction hypothesis does not fully explain pruning's effect on the generalization of VGG-16 model. In this section, we demonstrate that, with the exception of LeNet on MNIST, this phenomenon generalizes to other model architectures and pruning algorithms, thereby showing that size-reduction does not fully explain pruning's benefits to generalization in general.

**Method.**     For 5 model architecture and dataset combinations, we generate a family of models with different sparsities (or training epochs, in the case of the augmented training algorithm), using the following algorithms including 5 variants of pruning algorithms and the EDT algorithm:

1. Learning rate rewinding, as described in Section 2.

2. A variant of iterative magnitude pruning called *weight rewinding* [13, 53]. At the end of each pruning iteration, this algorithm rewinds not only the learning rate, but also values of remaining weights by resetting their values to the values they had had earlier in training.

3. Learning rate rewinding, but modified to remove weights according to a gradient-based weight selection criterion called *SynFlow* [60, 15]. This pruning algorithm uses the following heuristic to remove weights: the pruning algorithm first replaces each weight $w_l$ in the model with $\|w_l\|$; then, it feeds an input tensor filled with all 1's to this instrumented model, and the sum of the output logits is computed as $R$. This pruning algorithm then assigns an importance score $\|\frac{dR}{dw_l} \odot w_l\|$ to each weight, and remove the weights receiving the lowest such scores. Tanaka et al. [60] designed SynFlow to mitigate *layer collapse*, a phenomenon associated with ordinary magnitude-based pruning algorithm where weight removal concentrates on certain layers, effectively disconnecting the sparse model.

4. Learning rate rewinding, but modified to remove weights according to a gradient-based selection criterion called *SNIP* [34]. This pruning algorithm computes gradient $g_l$ for each layer $l$ using samples of training data. It then assigns an importance score $|w_l \odot g_l|$ to each weight, and removes the weights receiving the lowest such scores. The intuition behind this importance score is that it prevents pruning weights with the highest "effect on the loss (either positive or negative)" [15, 34].

5. *Iterative random* pruning, where a random set of weights are removed at each pruning iteration. The algorithm otherwise behaves like learning rate rewinding.

6. The *augmented* algorithm, which trains the model for the same number of epochs and with the same cyclic learning rate schedule as pruning, without removing weights.

In Table 2, for each benchmark and algorithm, we tabulate the test error of the model with the best validation error, selected from the family of models that each aforementioned algorithm generates. In Figure 8, we plot generalization of all models within the family of models each aforementioned algorithm generates as a function of sparsities and training time.

**Results.**     In Table 2, for each benchmark and algorithm, we select, from the family of models that each aforementioned algorithm generates, the model with the best validation error and tabulate its test error. In Figure 8, we plot the generalization of the family of models each aforementioned algorithm generates as a function of sparsities and training time in epochs. We observe that with the exception of LeNet on MNIST, the generalization of dense models that the augmented training algorithm produces matches or exceeds the generalization of sparse models that pruning algorithms produce. For LeNet on MNIST, however, the augmented training algorithm under-performs weight rewinding. We conjecture, without testing, that for this benchmark, the lack of any explicit form of regularization in the training process makes pruning's regularization effect uniquely important for generalization.

**Conclusion.**     In Section 1, We show that the augmented training algorithm produces VGG-16 models with generalization that is indistinguishable from that of models that pruning with learning rate rewinding produces. In this section, We further show that this phenomenon shows up even if

|  | M-LeNet | C10-ResNet20 | C10-VGG-16 | C100-ResNet32 | I-ResNet50 |
|---|---|---|---|---|---|
| LR Rewinding | 1.8±0.1 | 7.7±0.1 | 5.9±0.1 | 29.9±0.3 | 23.4±0.1 |
| Weight Rewinding | 1.5±0.1 | 8.1±0.3 | 6.2±0.1 | 30.6±0.4 | 23.5±0.1 |
| SynFlow | 1.7±0.1 | 7.8±0.3 | 6.1±0.1 | 30.2±0.2 | 23.9±0.1 |
| SNIP | 1.9±0.1 | 7.7±0.1 | 6.1±0.1 | 29.6±0.2 | 23.4±0.0 |
| Iterative Random | 1.9±0.2 | 8.1±0.3 | 6.4±0.2 | 30.2±0.1 | 23.9±0.1 |
| EDT | 1.8±0.1 | 7.6±0.3 | 6.0±0.1 | 29.0±0.1 | 23.4±0.3 |

Table 2: We tabulate test errors of the model with the minimum validation error that pruning and the EDT algorithm generate. With the exception of MNIST-LeNet, the EDT algorithm matches or exceeds the generalization of models that all pruning algorithms we test generate.

we apply other iterative pruning algorithms to additional model-dataset combinations. We therefore conclude that in general, size reduction does not fully account for pruning's benefits to generalization.

## D    Effects of Leaving Out Pruning-Affected Examples

In this section, we focus on the subgroups of examples whose training loss improves when pruning to a range of generalization-improving sparsities. By excluding them from the training dataset, we empirically examine their influence on the generalization of the dense models.

**Method.**    We refer to the top $K\%$ of training examples whose training loss improves the most during pruning as the *top-improved examples*. To examine the influence of these top-improved examples on generalization, for each sparsity pruning reaches, we train two dense models on two datasets respectively: a). the original training dataset excluding the top-improved examples at the specified sparsity, which we denote as TIE (**T**op-**I**mproved **E**xamples); b). a dataset of the same size as a)., but consisting of randomly drawn examples, which we denote as RND. We then compare their resulting generalization. If the top-improved examples are indeed the ones with the largest influence on generalization, excluding them from the training dataset should affect generalization more so than excluding a random subset of the same size. We set $K = 20$ in our experiments, we also tried setting $K = 10$, but the resulting generalization difference between models we train on two datasets is negligible.

**Result.**    We show in Figure 9 the generalization of dense models on the dataset excluding top-improved examples (TIE) and randomly drawn dataset of the same size (denoted RND) as a function of the sparsity at which the top-improved examples are selected. Figure 9 shows that, averaging within the sparsities where generalization improves (the range of sparsities in green), excluding top-improved examples hurts generalization more than excluding a random subset of examples of the same size by 0.78%, 0.77%, 0.34%, 0.69% for LeNet on MNIST, ResNet20 on CIFAR-10, VGG-16 on CIFAR-10, ResNet50 on ImageNet benchmarks. For ResNet32 on CIFAR-100 benchmark, we do not observe a significant difference between them. However, increasing the width of the ResNet32 model on CIFAR-100 from 16 to 128, we find that excluding the top-improved examples hurts generalization more than excluding a random subset of examples of the same size by 0.44% (c.f. the 1st image from the right on the 2nd row), similar to our finding on the remaining benchmarks.

**Conclusion**    The top-improved examples affect generalization to a greater extent than a randomly chosen set of examples of the same size. Moreover, on standard image classification benchmarks, they are more beneficial to generalization than randomly chosen examples.

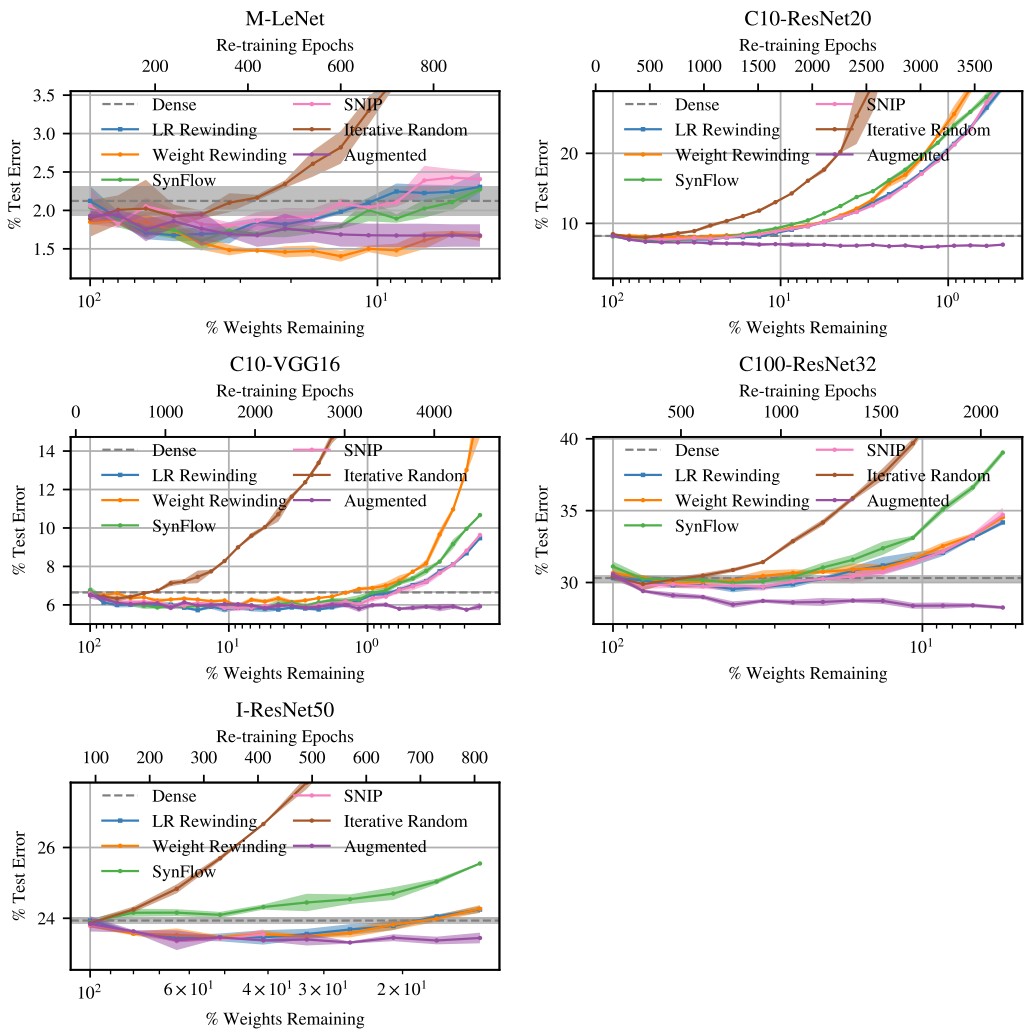

Figure 8: Comparing variants of pruning algorithm. With the exception of MNIST-LeNet benchmark, test errors of models we generate using the augmented pruning algorithm modified to no longer remove weights matches or exceeds the test errors of models we generate using all other variants of pruning algorithms.

# E    Leaving Out Noisy Examples Improves Generalization

Section 4 demonstrates that, when pruning to the optimal sparsity in the presence of random label noise, its effect is to increase training loss on noisy examples. In this section, we show that increasing training loss on the same set of noisy examples improves the generalization of dense models to the same, if not larger, extent as pruning. Our results validate the connection between increased training loss on noisy examples to generalization improvement.

**Method.**   Pruning to the optimal sparsity increases the training loss especially on a particular subgroup of examples consisting primarily of noisy examples. A simple and hyperparameter-free method to increase training loss of dense models on the same set of examples is to exclude from the training dataset the set of examples that the sparse models misclassify. Since training only on a dataset subset changes the number of gradient steps an epoch takes, we increase the total number of epochs so that the total number of gradient steps taken remains the same as training on the original dataset. We compare pruning and training dense models exclusively on the subset of examples that the sparse models correctly predict; we refer to such dense models as *dense-subset* models. In Figure 10,

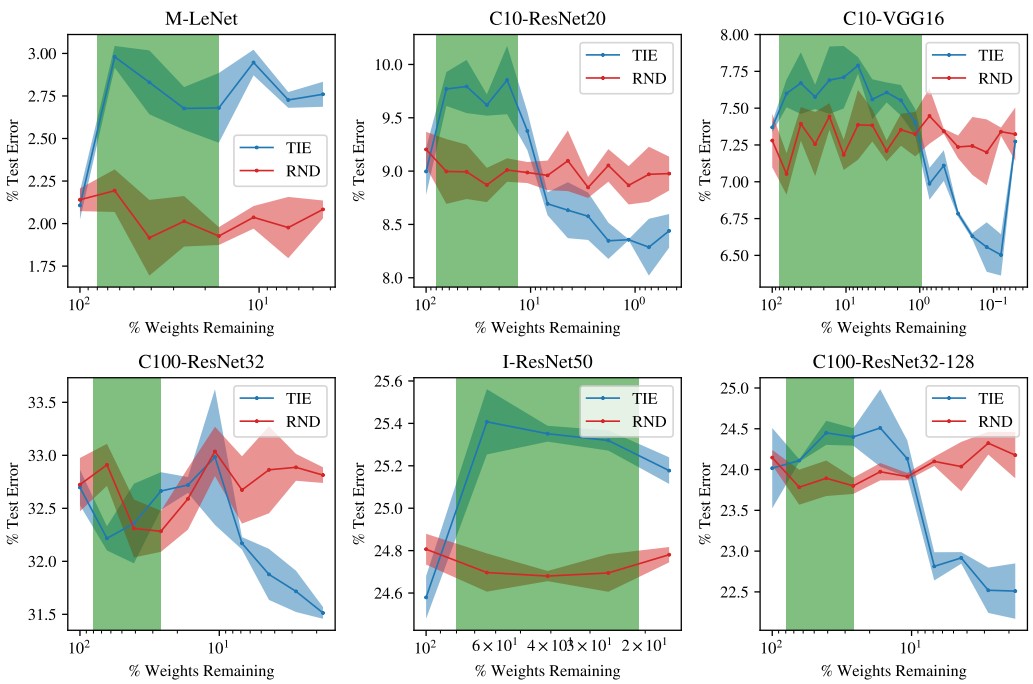

Figure 9: Excluding top-improved examples hurt generalization more than excluding a random subset of the same size on 4/5 benchmarks. We show test errors of dense models we train on the dataset excluding top-improved examples (denoted as TIE), and on the dataset excluding a random subset of the same size (denoted as RND) as a function of sparsity at which the top-improved examples are selected. The range of sparsities where generalization is improved by pruning is shaded in green.

plot the test errors of the dense-subset models as a function of the sparsity of the pruned model whose prediction determines which examples to exclude when training the said dense-subset model. For comparison, we also plot the test errors of sparse models as a function of its sparsity. We summarize the numerical results in Table 8.

**Results.** Variations in the generalization of the dense-subset models track variations in the generalization of the sparse models with respect to the fractions of weights remaining. Moreover, the generalization of the optimal dense-subset models matches or exceeds the generalization of the optimally sparse models.

**Conclusion.** Section 4 shows that, in the presence of random label noise, pruning to the optimal sparsity has regularization effects: pruning increases the loss on a select subgroup of training examples consisting predominantly of noisy examples. In this section, we demonstrate that increasing training loss on the same set of examples benefits dense model generalization as well. Our results establish a connection between pruning's regularization effect and generalization improvement.

# F Comparing Pruning with Width Down-scaling

In this section, we present a less-cluttered version of the images we show in Section 5.2.

# I Similarities between Pruning and Width Down-scaling

In Section 5.2, we observe that the pruned models and width down-scaled models attain similar generalization at the optimal sparsity and model width on benchmarks with random label noise, where pruning's effect at the optimal sparsity is to improve generalization by strengthening regularization (Section 4). Their similarity is surprising because pruning and down-scaling model width are two

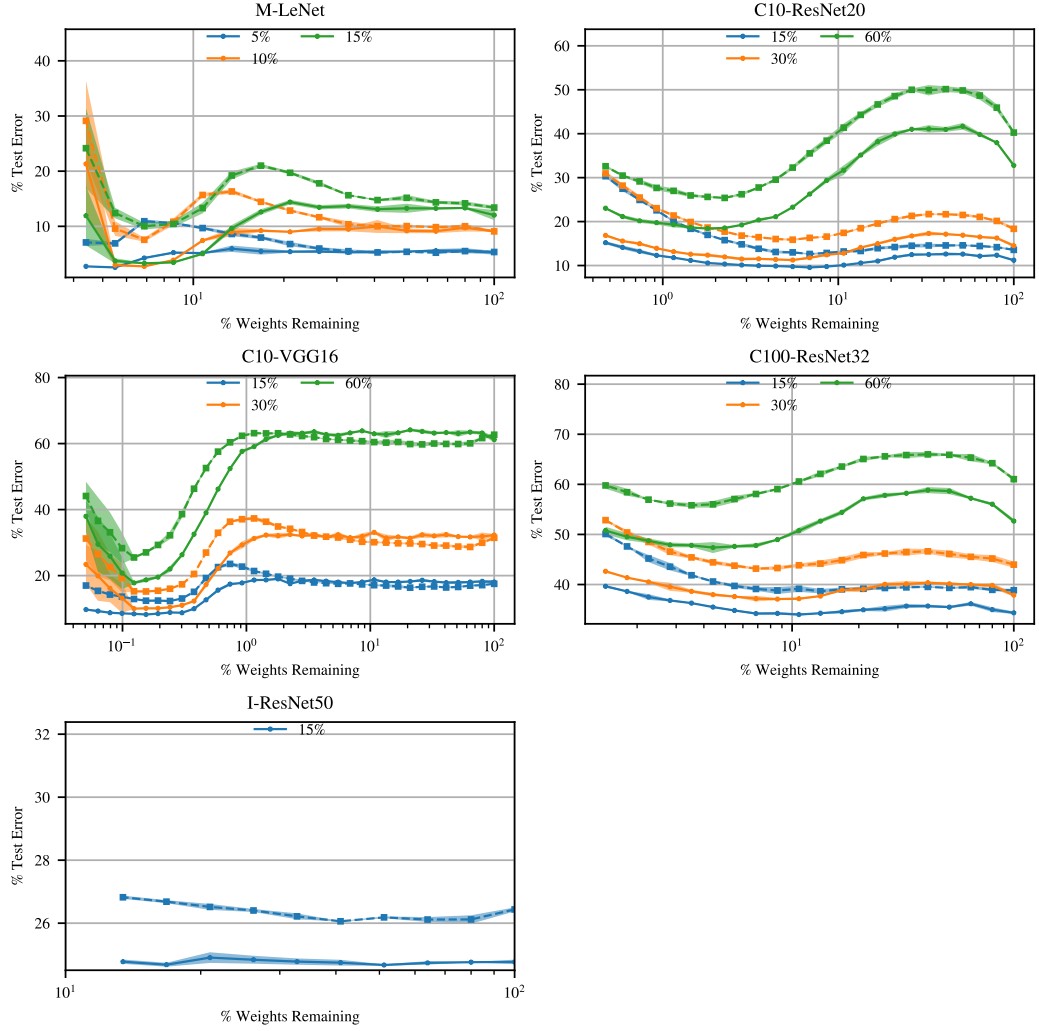

Figure 10: Excluding examples that the optimally sparse models misclassify improves dense model generalization to the same if not greater extent as pruning does. Test errors of dense-subset models are shown using circular markers with solid lines. Test errors of sparse models are shown using square markers with dashed lines. Legend shows levels of random label noise.

distinct methods to reduce model size. In this section, we further compare pruning and width down-scaling with equalized training time and quantify the similarities between their effects on training examples.

**Method.** Width down-scaling differs from pruning because a pruned model receives extra training, due to the retraining step. To compare fairly, we modify the pruning and width down-scaling algorithms as such to equalize the amount of training:[7]

1. For pruning, we adopt a variant of the magnitude pruning algorithm, called *one-shot pruning* [53]. While iterative pruning (the pruning algorithm we introduce in Section 2) removes a fixed fraction of the remaining weights per iteration until achieving the desired sparsity,

---

[7]Alternatively, one can equalize training by increasing the amount of training of width down-scaled models. However this increases the amount of compute required by an order of magnitude, and beyond reasonable budget. This is because unlike iterative pruning, a trained model with a larger width does not become the starting point for training a model with a smaller width, and therefore no amount of training is shared between models with different sizes.

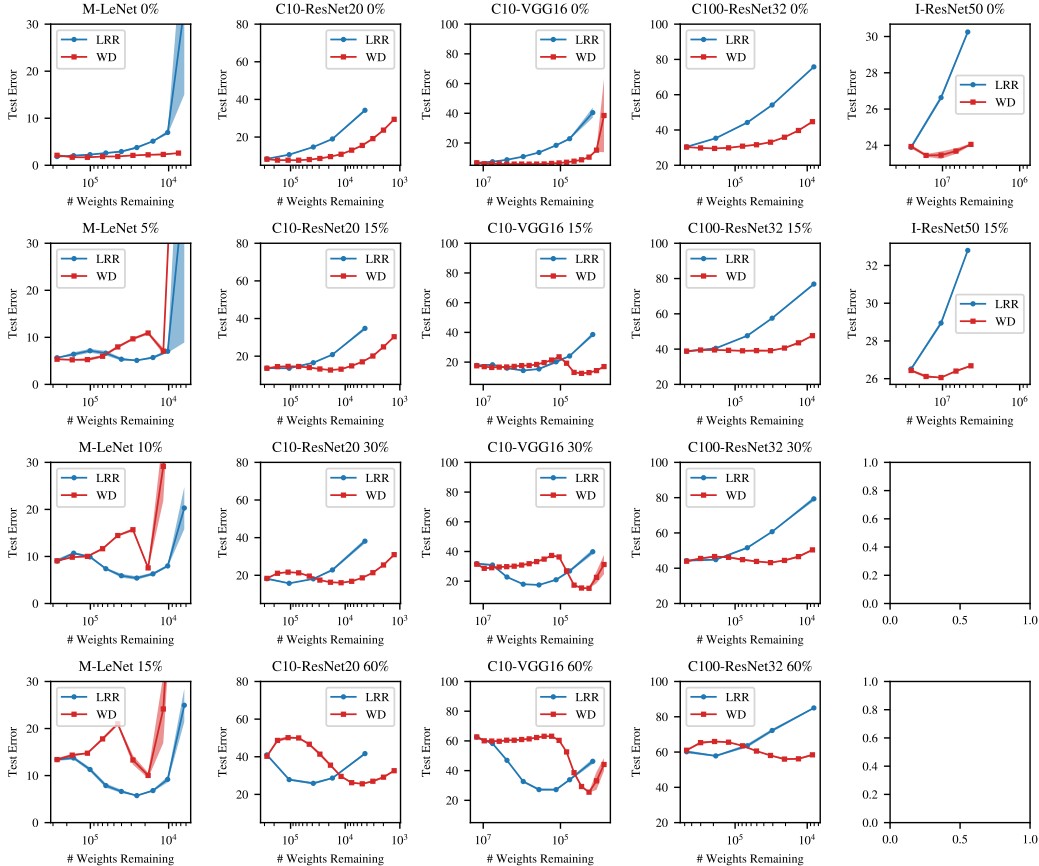

Figure 11: Comparing Pruning with Model Width Reduction. The percentage of random label noise injected is shown in each subplot title. The test errors of sparse models generated by learning rate rewinding algorithm (denoted LRR) and width-down-scaled models (denoted WD) as a function of their number of weights are shown using orange and blue lines, respectively.

one-shot pruning removes all weights at once to reach the desired sparsity. Like iterative pruning, one-shot pruning retrains the model after weights removal.

2. For width down-scaling, we equalize the amount of training with one-shot pruning by training the width down-scaled model for the same number of gradient steps and using the same learning rate schedule as one-shot pruning.

We then quantify the similarities between two algorithms by comparing (1). the generalization of their produced models; (2). average loss change, relative to the standard dense model, on training example subgroups with distinct EL2N percentile ranges due to respective algorithms; (3). overlap of the 10% examples whose training loss changes the most due to two algorithms, relative to the training loss of the original dense model, measured in terms of *Jiccard similarity index*. The Jiccard similarity index $J(A, B)$ measures the similarity between the content of two sets $A$ and $B$, with the following formula:

$$J(A, B) = \frac{|A \cap B|}{|A \cup B|}$$

**Generalization.** Figure 12 shows that generalization of models that both algorithms produce are qualitatively similar on benchmarks with injected random label noise – generalization of models that both algorithms generate improves when they only remove a relatively small fraction of weights. As both algorithms remove more weights, generalization of produced models begins to suffer.

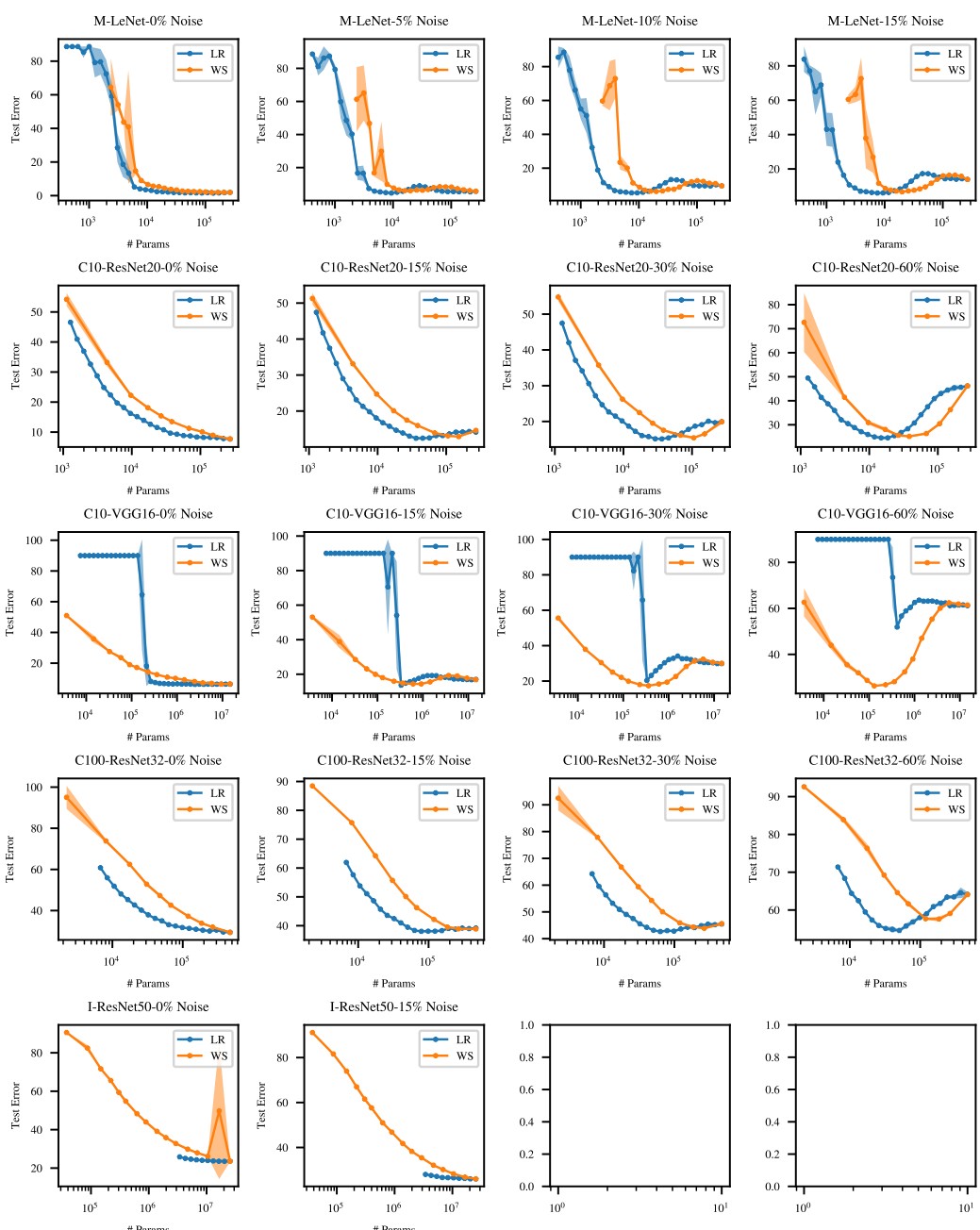

Figure 12: Generalization of pruned models (denoted LR) is similar to generalization of width down-scaled models (denoted WS). We equalize training time between pruning and width down-scaling.

**Change in loss by subgroups.** Figure 13 shows that on benchmarks with random label noise, pruning and width down-scaling has similar effects on example subgroups with distinct EL2N score percentile ranges at optimal sparsity and model width respectively – training loss increase for both algorithms tends to concentrate on examples with high EL2N score percentile ranges.

**Affected examples overlap.** Table 3 shows that the 10% of training examples whose training loss changes the most due to pruning and width down-scaling overlap significantly more than random baseline.

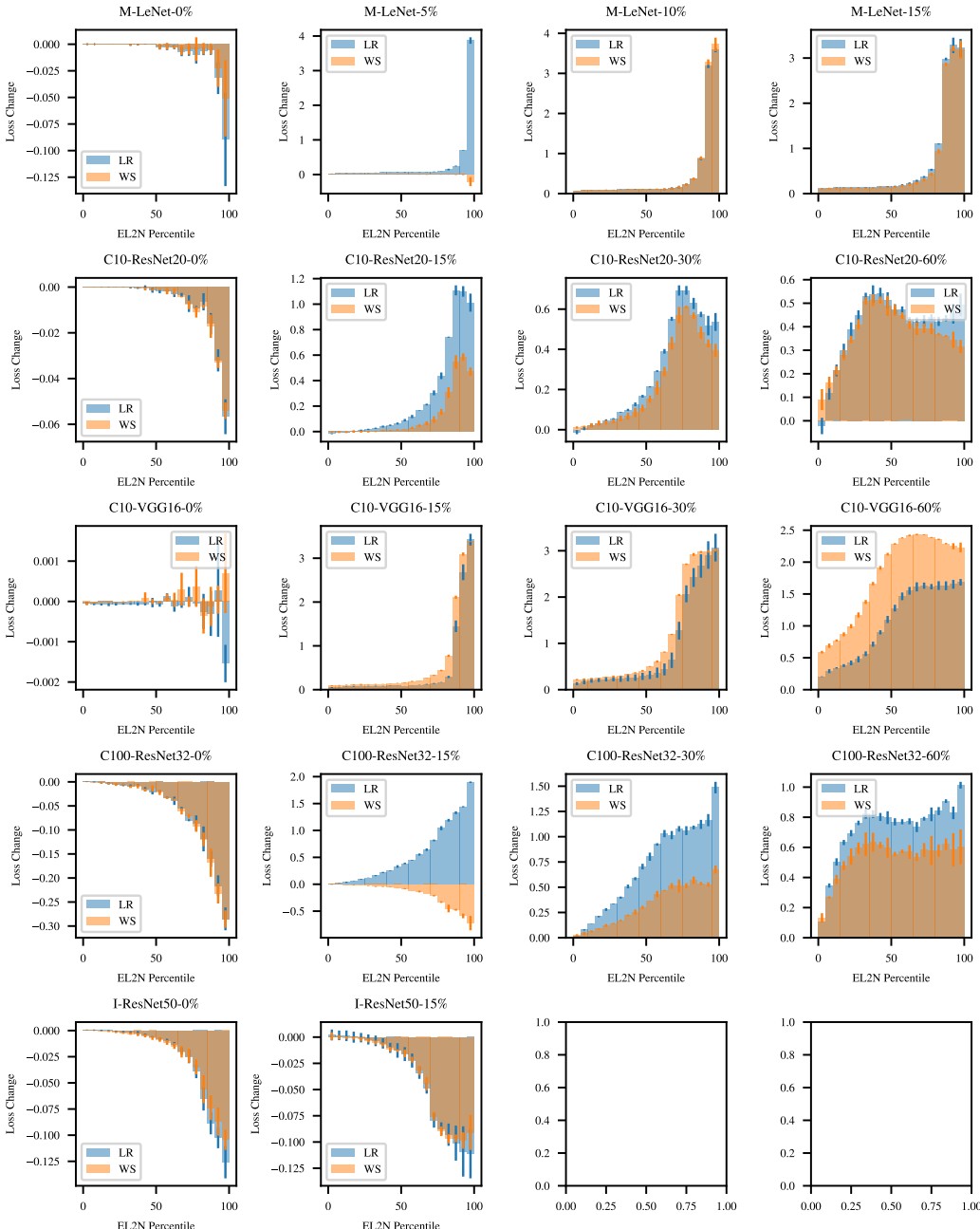

Figure 13: Pruning (denoted LR) and width down-scaling (denoted WS) has similar effect on the average training loss of example subgroups with distinct EL2N score percentile ranges at optimal sparsity and model width.

| Model | Noise Level | Jaccard Similarity |
|---|---|---|
| M-LeNet | 0%/5%/10%/15% | 0.60 / 0.11 / 0.74 / 0.62 |
| C10-ResNet20 | 0%/15%/30%/60% | 0.20 / 0.37 / 0.35 / 0.42 |
| C10-VGG-16 | 0%/15%/30%/60% | 0.14 / 0.57 / 0.27 / 0.12 |
| C100-ResNet32 | 0%/15%/30%/60% | 0.17 / 0.23 / 0.27 / 0.28 |
| I-ResNet50 | 0%/15%/ | 0.26 / 0.26 |

Table 3: Jaccard similarity of top 10% of examples most affected by pruning and width down-scaling. A randomly chosen two sets of 10% training examples has a baseline similarity index of 0.05. All standard deviations are less than or equal to 0.03.

**Conclusion.** Pruning and width down-scaling have similar regularization effects: they nonuniformly increase the average training loss of example subgroups, which we show is key to improving generalization in the presence of random label noise (Section 4). Therefore, the regularization effect of pruning may be a consequence of model size reduction in general.

## J    Ablation Studies

Pruning algorithms are complex aggregation of individual design choices. In this section, we examine whether specific design choices of pruning algorithms including weight resetting and weight selection heuristics contributes to the generalization-improving effect of pruning.

### J.1    Rewinding Weights

The pruning algorithm we use in this study, namely, learning rate rewinding, rewinds learning rate to their value early in training after each pruning iteration. Its predecessor, an iterative pruning algorithm called weight rewinding rewinds not just learning rate, but also weights to their values early in training after each pruning iteration. Frankle and Carbin [13] proposed weight rewinding and later, Renda et al. [53] found that rewinding weight is unnecessary if one's goal is to produce a family of models achieving the optimal parameter count - generalization trade-off. In this section, we expand the comparison weight rewinding [13] and learning rate rewinding [53] to datasets with random label noise, and show that rewinding weights is actually necessary for achieving optimal generalization on benchmarks with random label noise.

**Method.** To examine whether rewinding weights contributes to the generalization-improving effect of pruning, we compare the generalization of models that weight rewinding produces with generalization of models that learning rate rewinding produces. The former pruning algorithm rewinds weights and the latter does not.

**Results.** We plot our results in Figure 14. Numerical results are available in Table 9 in Appendix M. We observe that across 18 model, dataset, noise level combinations, weight rewinding out-performs learning rate rewinding on 10 benchmarks, whereas learning rate rewinding only out-performs weight rewinding on 2. For the remaining 6 benchmarks, two algorithms produce models with matching generalization. Notably, weight rewinding is better at mitigating random label noise than learning rate rewinding. Across all model architectures, on datasets with and without random label noise, the optimal test error of models that weight rewinding produces is lower than the optimal test error of models that learning rate rewinding produces by -0.7% to 0.3%, 0% (matching) to 5.5%.

**Conclusion.** Weight resetting contributes to the generalization-improving effect of pruning.

### J.2    Weight Selection Heuristics

The pruning algorithm we study, namely learning rate rewinding [53], removes weights based on the simple heuristic that weights with low magnitude are less important for the given task. We study whether pruning's generalization improving effect is affected by the choice of weight selection heuristic.

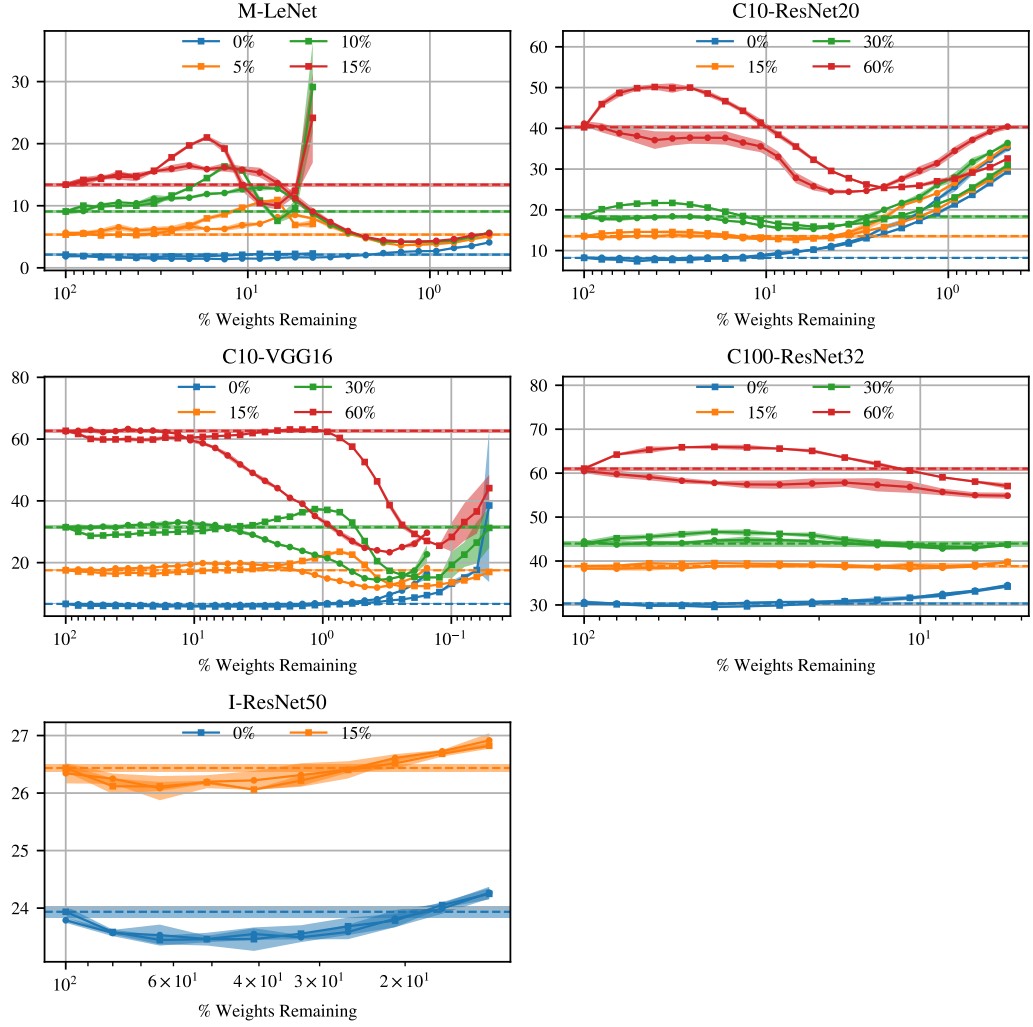

Figure 14: Rewinding weights helps improve generalization in the presence of random label noise. We plot test errors of models that weight rewinding and learning rate rewinding produce in lines with circle and square markers, respectively. We plot test errors of dense models with horizontal dashed lines. Legend shows noise levels.

**Method.** We modify learning rate rewinding to adopt the following alternative weight selection heuristics, and compare the generalization of models that each modified pruning algorithm generates.

1. Random selection. Weights are removed at random without considering their values.
2. Synaptic flow preserving weight selection [60]. This weight selection heuristic works as follow. The heuristic algorithm first replaces each weight $w_l$ in the model with $\|w_l\|$; then, the algorithm feed an input tensor filled with all 1's to this instrumented model, and the sum of the output logits is computed as $R$. The heuristic algorithm then assigns an importance score $\|\frac{dR}{dw_l} \odot w_l\|$ to each weight, and the weights receiving the lowest such scores are removed. Tanaka et al. [60] designed SynFlow to mitigate *layer collapse*, a phenomenon associated with ordinary magnitude-based pruning method where weight removal concentrates on certain layers, effectively disconnecting the sparse model.

**Results.** We plot and tabulate the result of comparison in Figure 15 and Table 10. We observe that SynFlow-based weight selection heuristic beats magnitude-based one on 6 out of 18 benchmarks. We also observe that random pruning is particularly effective at mitigating random label noise – on 9 out

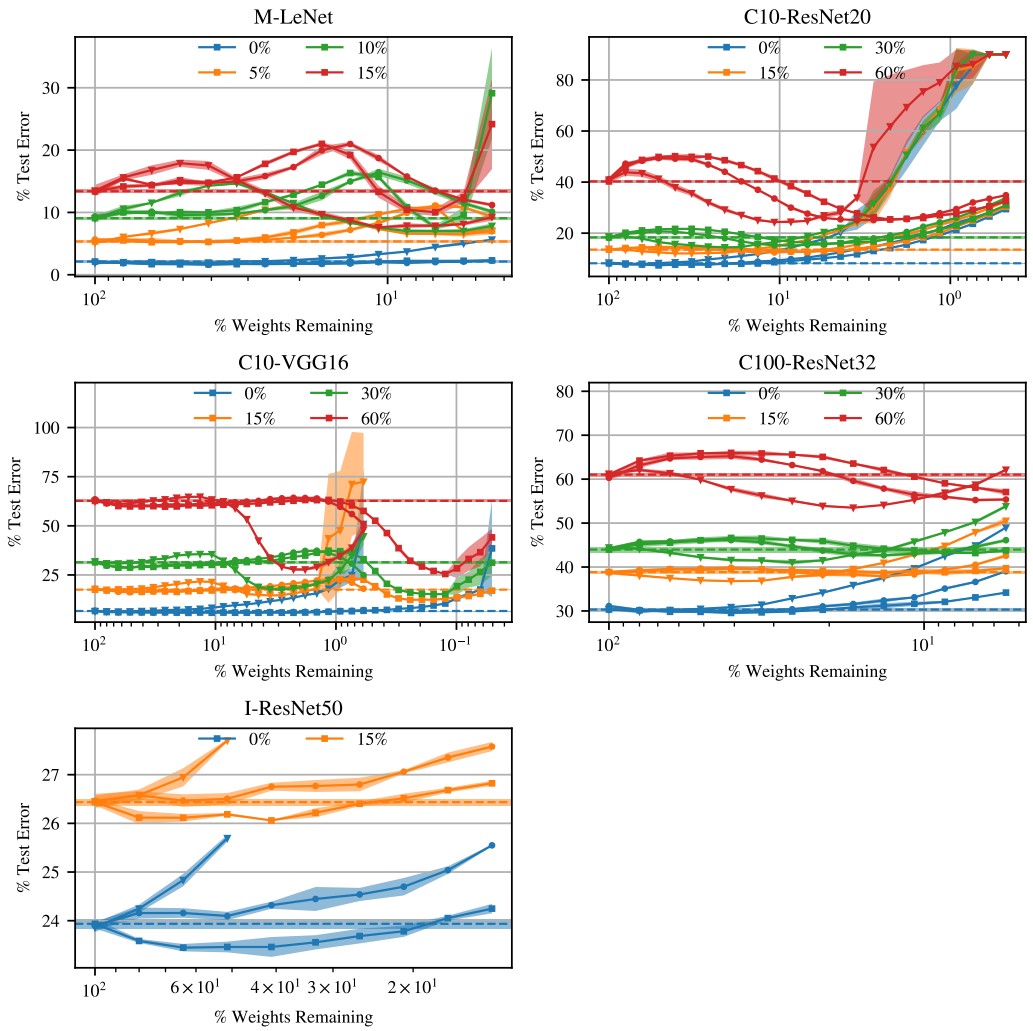

Figure 15: The choice of weight selection heuristic matters to generalization. However, among magnitude, synflow and random selection heuristics we test, not one selection strategy stands out as the best heuristic for improving generalization. We plot test errors of models that pruning with magnitude, synflow and random selection heuristics produce in lines with square, circle and triangle markers, respectively. We plot test errors of dense models with horizontal dashed lines. Legend shows noise levels.

of 13 benchmarks with random label noise, random pruning out-performs magnitude-based weight selection heuristic.

**Conclusion.** The choice of weight selection heuristic plays a significant role in determining pruning's generalization-improving effect.

# K   Closer Look at Examples with Training Loss Most Worsened by Pruning

In this section, we take a closer look at the examples in the standard image classification training datasets whose training loss is increased the most by pruning to a range of generalization-improving and generalization-preserving sparsities.

| | MNIST LeNet | CIFAR-10 ResNet20 | CIFAR-10 VGG-16 | CIFAR-100 ResNet32 | ImageNet ResNet50 |
|---|---|---|---|---|---|
| Most-worsened | 1.4% | 16.7-13.4% | 2.3-0.7% | 16.7-8.6% | 13.4% |
| Most-improved | 80-5.5% | 80-21% | 80-2.3% | 80-21% | 80-16.8% |
| Least-affected | 100-1.4% | 100-5.5% | 100-0.19% | 100-8.6% | 100-13.4% |

Table 4: Sparsities at which we measure training loss after pruning to select training dataset subsets.

### K.1   Example Images and Labels

When pruning to sparsities that improve or preserve (causing <2% error increase) generalization, what is the nature of examples whose training loss increases or decreases the most? Here, we present examples whose training loss is affected the most by pruning to the aforementioned sparsities. We show that while the majority of these examples are atypical representations of their labels, a fraction of these examples contain incorrect or ambiguous labels.

**Method.**   We select three subsets of training examples: the most-worsened examples, the most-improved examples and the least-affected examples as follow.

1. *The most-worsened examples:* we first measure the per-example training loss after pruning to the following sparsities. For ResNet20 on CIFAR-10 and VGG-16 on CIFAR-10 benchmarks, a range of sparsities with an increased overall training loss but improved generalization exist. We measure per-example training loss after pruning to these sparsities. For LeNet on MNIST, ResNet32 on CIFAR-100 and ResNet50 on ImageNet benchmarks, however, no such sparsities exist. We therefore choose to measure per-example training loss after pruning to sparsities with an increased overall training loss and generalization that is no worse than that of the dense models by 2%. We then rank examples in the training dataset based on the geometric average of their training loss increase after pruning to aforementioned sparsities, relative to their training loss before pruning.[8] We select examples with the most training loss increase as the most-worsened examples.

2. *The most-improved examples:* we first measure the per-example training loss after pruning to sparsities with improved generalization and decreased training loss. We then rank examples in the training dataset based on the geometric average of their training loss decrease after pruning to aforementioned sparsities, relative to their training loss before pruning.[9] We select examples with the most training loss decrease as the most-improved examples.

3. *The least-affected examples:* we first measure the per-example training loss after pruning to sparsities that attains generalization no worse than that of the dense models by 2%. We then rank examples in the training dataset based on geometric average of the absolute value of their training loss change after pruning to aforementioned sparsities, relative to their training loss before pruning. We select examples with the least absolute value of training loss change as the least-affected examples.

For each type of training example subset, we tabulate the sparsities at which we measure per-example training loss after pruning in Table 4. We average the per-example training loss across selected sparsities and across independent runs of the same experiment with distinct random seeds.

On each benchmark, we present the 20 examples from each category Figure 16. Notably, the set of 20 least-affected examples are not unique, as many examples are practically unaffected by pruning.

**Results.**   We describe our observations in the captions of Figure 16.

---

[8]Since we are concerned with examples whose training loss increases due to pruning, and in general, one cannot compute geometric average of arrays with negative numbers, we set negative training loss increase due to pruning to a small number $1e-5$, effectively ignoring them.

[9]Similar to how we select most-worsened examples, since we are concerned with examples whose training loss decreases due to pruning, and in general, one cannot compute geometric average of arrays with negative numbers, we set negative training loss decrease due to pruning to a small number $1e-5$, effectively ignoring them.

**Conclusion.** When pruning to sparsities that improves or preserves generalization (<2% error increase), the set of examples whose training loss is most affected (i.e., either most worsened or most improved) are examples that are atypical representations of their labels. A fraction of them have ambiguous or erroneous labels. In contrast, the least affected examples are mostly unambiguous and canonical representations of the labels.

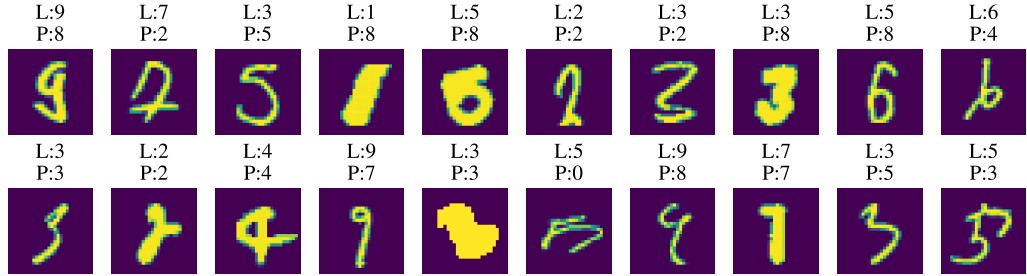

(a) MNIST-LeNet Most-worsened Examples. The 3rd image on the 1st row has the incorrect label (should be 5, but labeled 3). The 2nd image on the 1st row is labeled 7, but is indistinguishable from a 2. The 5th image on the 1st row is labeled 5, but it also looks like an 8.

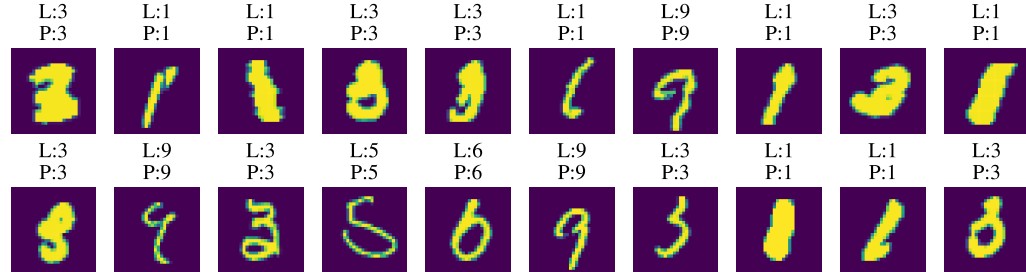

(b) MNIST-LeNet Most-improved Examples. Similar to the most-worsened examples, such examples are atypical representation of labels, and may contain wrong or ambiguous labels.

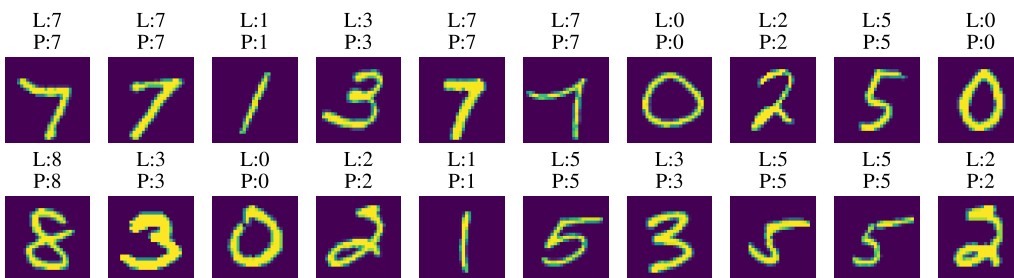

(c) MNIST-LeNet Least-affected Examples.

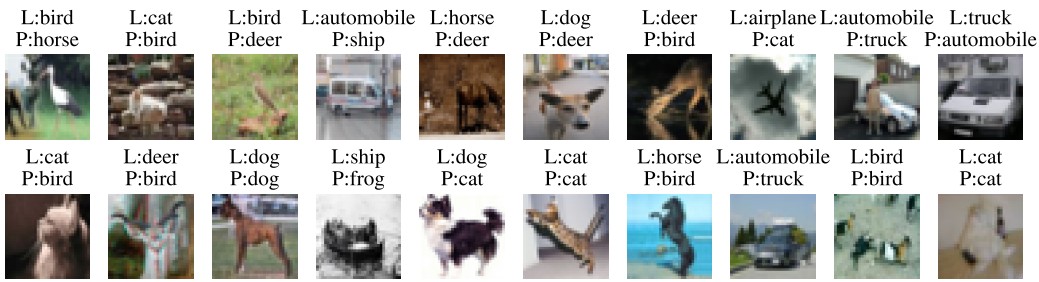

(d) CIFAR-ResNet20 Most-worsened Examples. The 1st image on the 1st row, the 5th image on the 1st row and the 2nd image from the right on the 1st row contains multiple objects with the same label. The 1st image from the right on the 1st row is labeled truck, but cannot be discerned from an automobile when only its front is shown.

Figure 16: Most-worsened/improved examples are atypical representations of labels and may have wrong and ambiguous labels. Least-affected examples are unambiguous and canonical representations of labels. Plot title shows image label (denoted L) and model majority prediction (P).

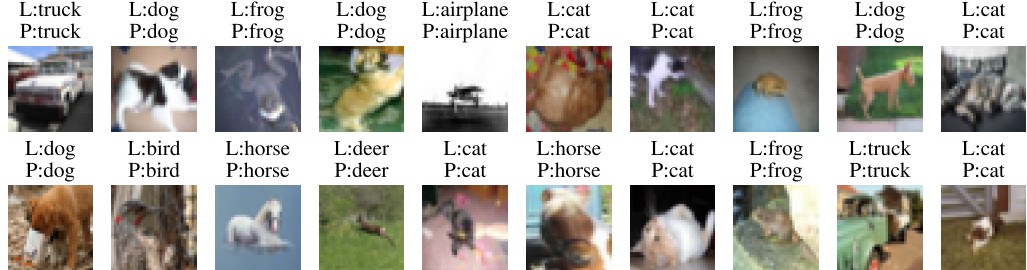

(e) CIFAR-ResNet20 Most-improved Examples. Similar to the most-worsened examples, such examples are atypical representation of labels and may contain wrong or ambiguous labels.

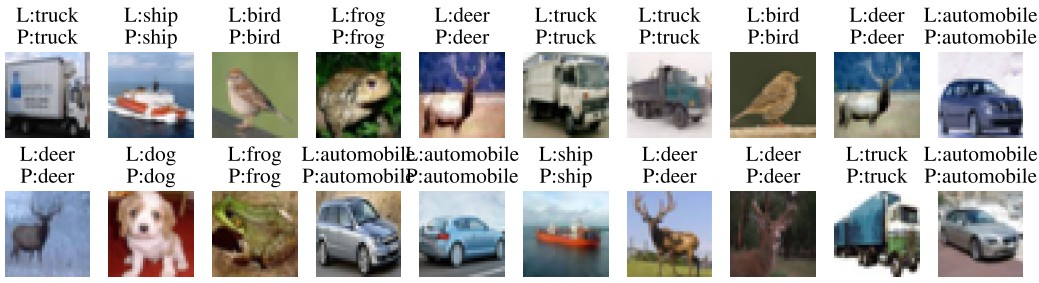

(f) CIFAR-ResNet20 Least-affected Examples.

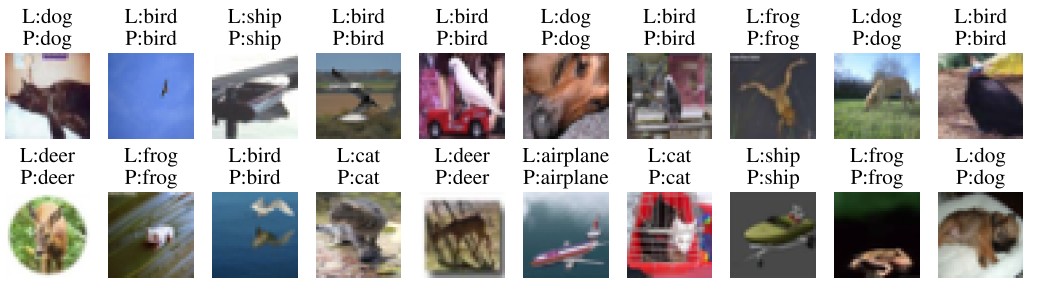

(g) CIFAR-VGG-16 Most-affected Examples. The 5th image on the 1st row contains multiple objects with the same label. The 2nd image on the 1st row labeled bird does not contain enough information to be discerned from an airplane. The 2nd image from the right on the 1st row is quite blurred, but judging by the fact that horses are more likely to graze than dogs, the animal in this picture is likely to be a horse.

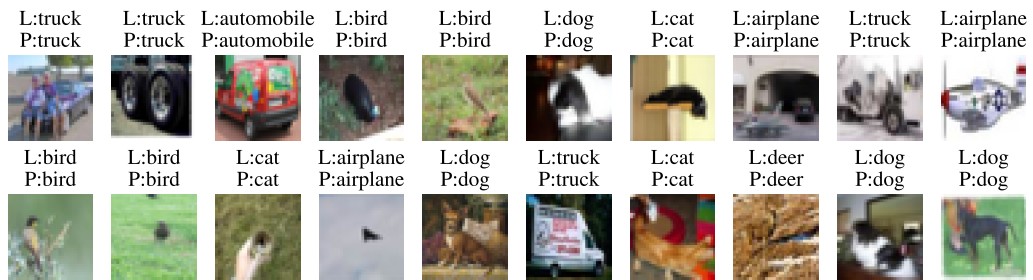

(h) CIFAR-VGG-16 Most-improved Examples. Similar to the most-worsened examples, such examples are atypical representation of labels and may contain wrong or ambiguous labels..

Figure 16: (Cont.) Most-worsened/improved examples are atypical representations of labels and may have wrong and ambiguous labels. Least-affected examples are unambiguous and canonical representations of labels. Plot title shows image label (denoted L) and model majority prediction (P).

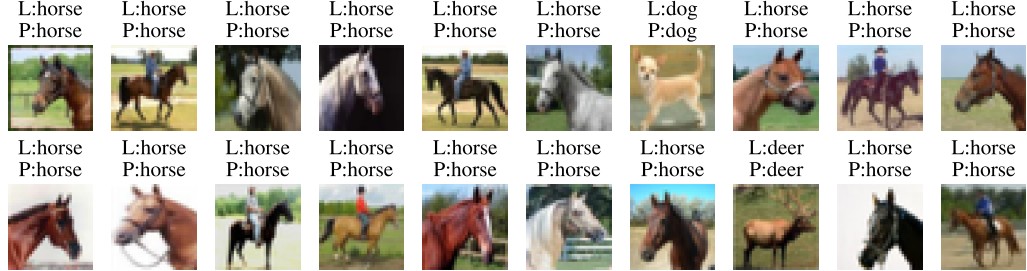

(i) CIFAR-VGG-16 Least-affected Examples.

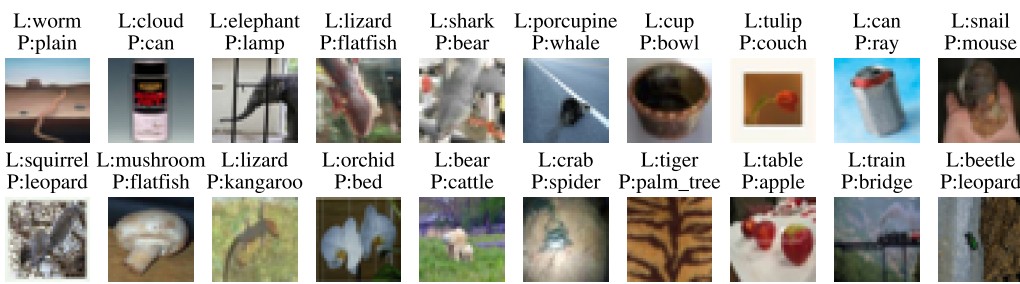

(j) CIFAR-100-ResNet32 Most-affected Examples. The 2nd image on the first row labeled cloud has the wrong label. The 3rd image from the right on the 2nd row labeled table has more than one objects with the same label, it also has fruits on it.

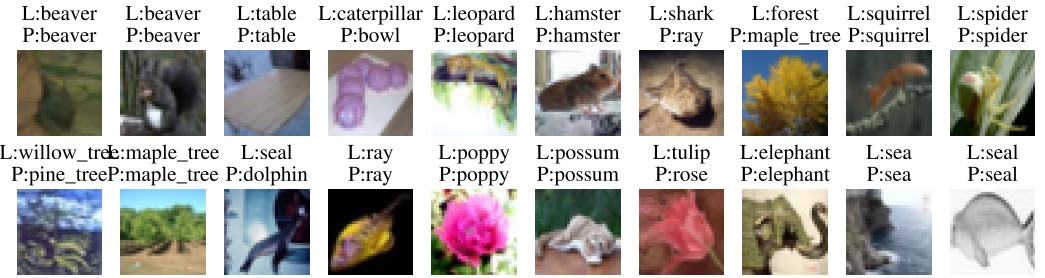

(k) CIFAR-100-ResNet32 Most-improved Examples. Similar to the most-worsened examples, such examples are atypical representation of labels and may contain wrong or ambiguous labels.

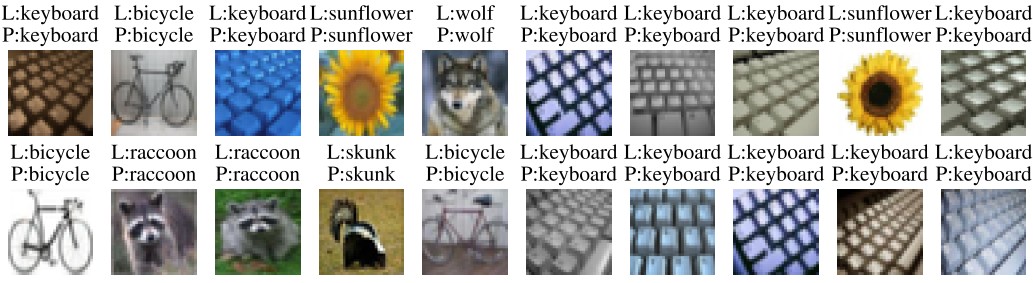

(l) CIFAR-100-ResNet32 Least-affected Examples.

Figure 16: (Cont.) Most-worsened/improved examples are atypical representations of labels and may have wrong and ambiguous labels. Least-affected examples are unambiguous and canonical representations of labels. Plot title shows image label (denoted L) and model majority prediction (P).

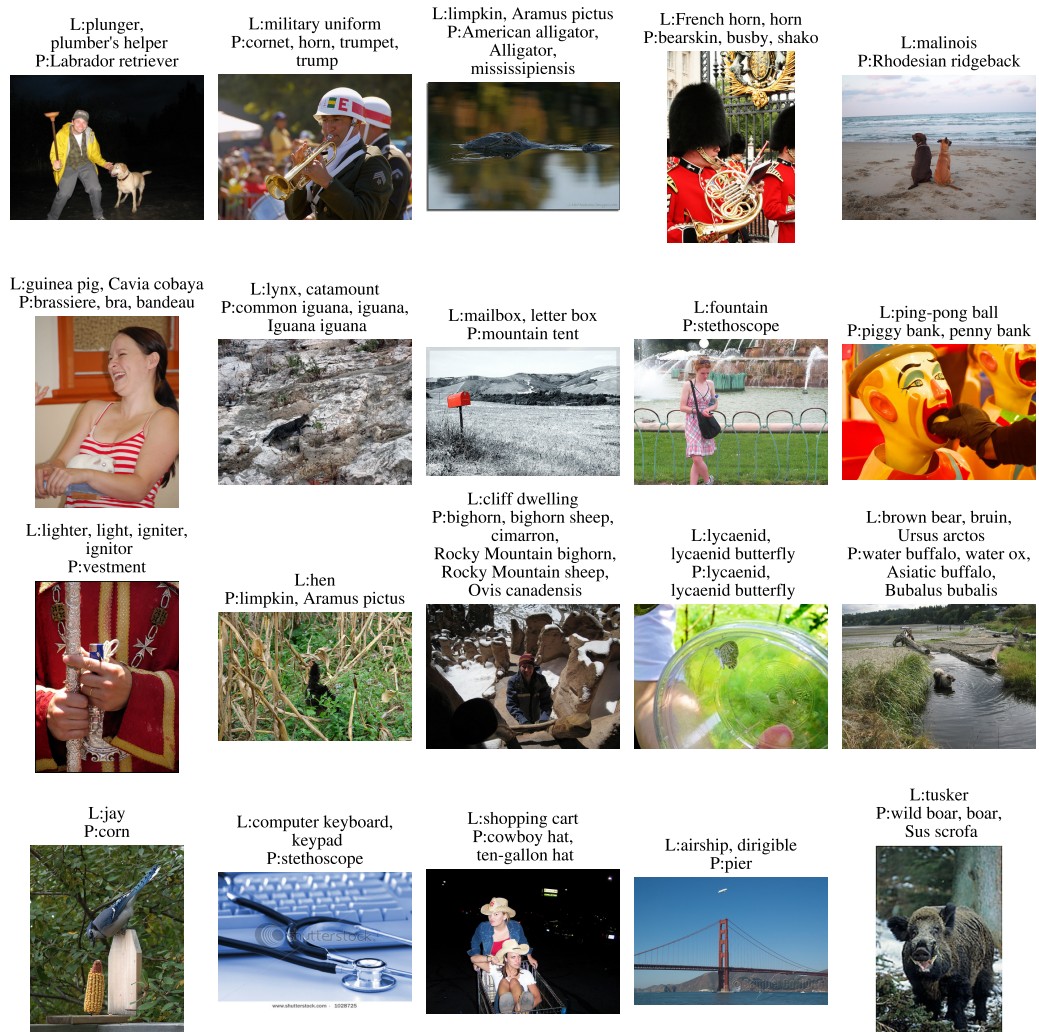

(m) ImageNet-ResNet50 Most-affected Examples. The 3rd image on the 1st row is not a limpkin, which is a bird species with long beak. The 1st image from the right on the 2nd row is not a tusker, but a wild boar. The 5th image from the right labeled guinea pig, the 2nd, 3rd and 4th images from the right on the 2nd row, among other images, show multiple objects with the same label.

Figure 16: (Cont.) Most-worsened/improved examples are atypical representations of labels and may have wrong and ambiguous labels. Least-affected examples are unambiguous and canonical representations of labels. Plot title shows image label (denoted L) and model majority prediction (P).

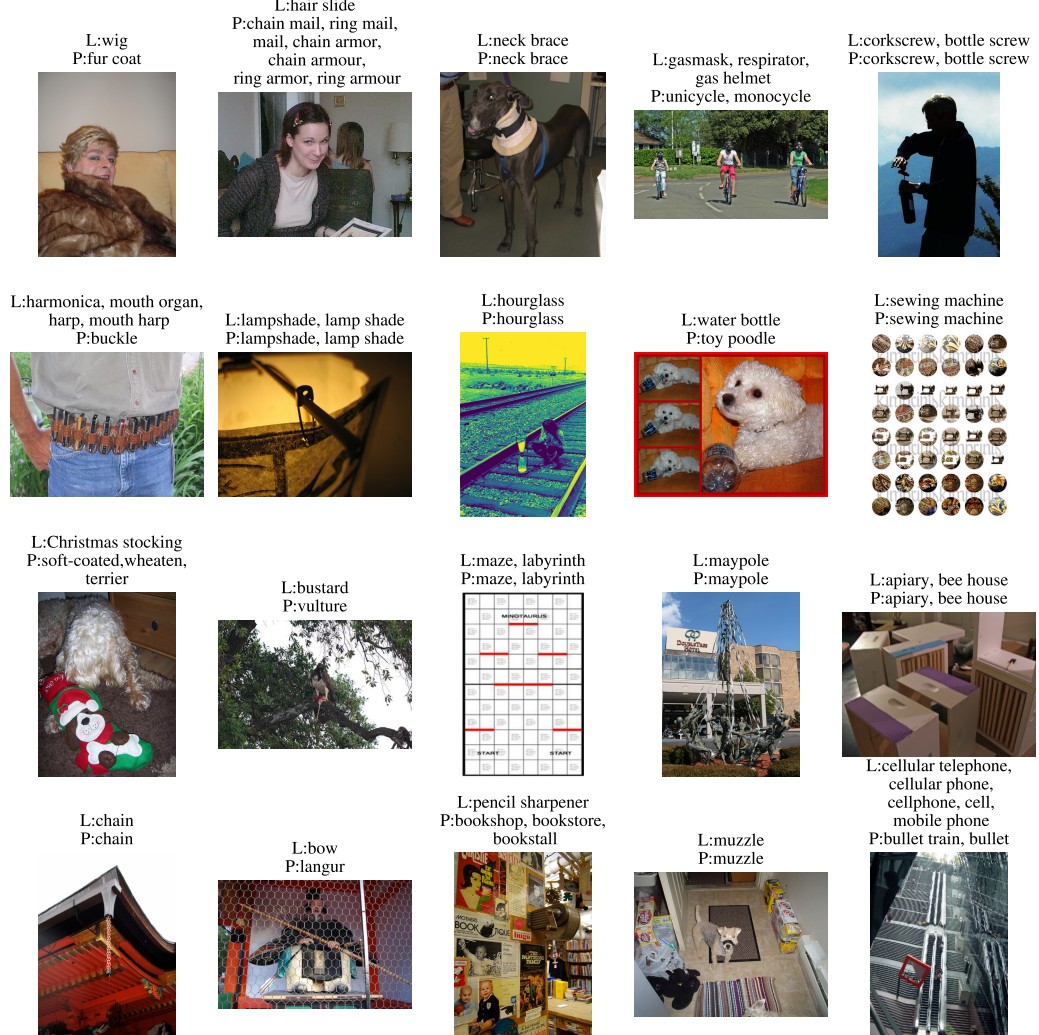

(n) ImageNet-ResNet50 Most-improved Examples. Similar to the most-worsened examples, such examples are atypical representation of labels and may contain wrong or ambiguous labels.

Figure 16: (Cont.) Most-worsened/improved examples are atypical representations of labels and may have wrong and ambiguous labels. Least-affected examples are unambiguous and canonical representations of labels. Plot title shows image label (denoted L) and model majority prediction (P).

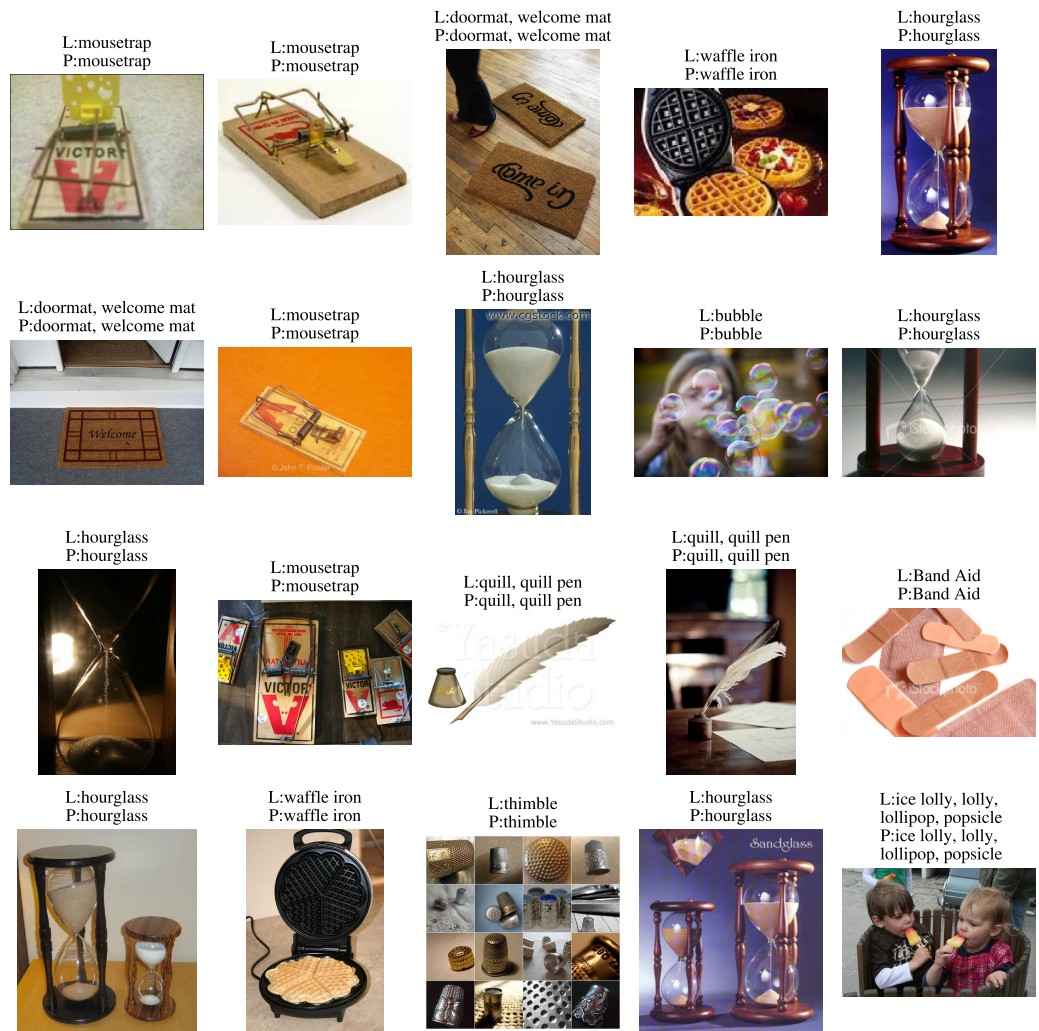

(o) ImageNet-ResNet50 Least-affected Examples.

Figure 16: (Cont.) Most-worsened/improved examples are atypical representations of labels and may have wrong and ambiguous labels. Least-affected examples are unambiguous and canonical representations of labels. Plot title shows image label (denoted L) and model majority prediction (P).

### K.2 Empirical Analysis

We refer to the set of examples whose training loss is increased the most by pruning as *top-worsened examples*. In this subsection, we empirically evaluate the effect of the top-worsened examples on generalization. We focus on two benchmarks – CIFAR-10-ResNet20 and CIFAR-10-VGG-16, since we only observe generalization improvement with an overall *increase* in training loss on the two aforementioned benchmarks.

**Method.**  For each benchmark and model with sparsity $X\%$ that the pruning algorithm generates, we create two dataset subsamples of size $N$ to evaluate the effect of $P$ top-worsened examples on generalization, where $N \gg P$: a). start with a randomly drawn $(N-P)$ examples, and add additional $P$ examples whose training loss is increased the most by pruning to sparsity $X$ (i.e., the top-worsened examples); we denote this dataset subsample as $S_{TW}^X$; b). start with the same $(N-P)$ examples, but add additional $P$ examples drawn randomly from the rest of the dataset; we denote this dataset subsample as $S_{Rand}^X$. We then train dense models on two subsamples $S_{TW}^X$ and $S_{Rand}^X$ to obtain the corresponding generalization $Y_{TW}^X, Y_{Rand}^X$. If our hypothesis is correct – that pruning improves generalization by increasing training loss, essentially ignoring, a small fraction of noisy examples detrimental to generalization – for a range of sparsities $X$ where pruning improves generalization, we should observe that the generalization of dense models we train on the subsample containing top-worsened examples ($S_{TW}^X$) to be worse than the one we train on a randomly drawn subsample ($S_{Rand}^X$). Following the precedent of Paul et al. [51], we choose $N$ to be the size equivalent to $40\%$ of the training dataset and $P$ to be $1\%$ of the training dataset.

Notably, subsampling the dataset is necessary to reveal the effect of the small fraction of noisy examples empirically. Their relatively rare occurrence makes their harmful effect on generalization difficult to detect empirically. This is consistent with with observation and experimental setup used in Paul et al. [51], characterizing the harmful effect of a small fraction of training examples with high EL2N scores. We similarly demonstrate the harmful effect of examples avoided by the sparse models.

**Results.**  We visualize the generalization difference $Y_{TW}^X$ - $Y_{Rand}^X$ of dense models we train on the two dataset subsamples as a function of sparsities $X$ in Figure 17. Indeed, dense models we train on the subsample containing the top-worsened examples achieve worse generalization compared with dense models we train on a random subsample, most notably near and beyond the highest sparsity that still improves generalization. Within the range of sparsity levels where pruning improves generalization, the test errors of dense ResNet20, VGG-16 models we train on the dataset subsample containing the top-worsened examples is worse than the models trained on a random subsample by 0 (matching) to 0.35% and 0 (matching) to 0.54% respectively. We thus confirm that pruning improves generalization while avoiding fitting a small fraction of examples harmful to generalization.

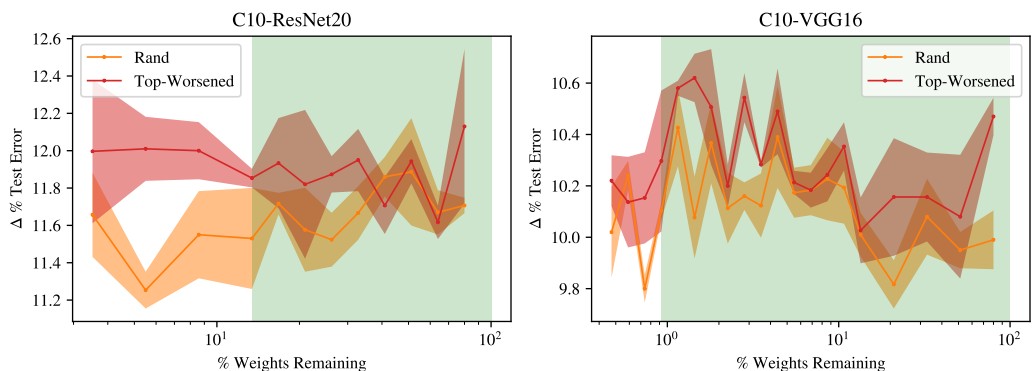

Figure 17: Dense models we train on dataset subsamples including top-worsened examples (labeled Top-Worsened) achieve worse generalization than dense models we train on dataset subsamples without top-worsened examples (labeled Rand). We show test errors of dense models trained on two dataset subsamples as a function of sparsities at which per-example training loss is measured. We color the range of sparsity levels where pruning improves generalization green.

# L    Effects of Learning Algorithms on Subgroup Training Loss

In Section 3, we partition the training set into subgroups, each with a distinct EL2N score percentile range and present pruning's effect on the average training loss of example subgroups at two sparsities of interests. In this section, we present pruning's effect on the average training loss of example subgroups at more sparsities. We similarly present the effect of two other learning algorithms, extended dense training (Section 5.1) and width downscaling (Section 5.2), on subgroup training loss.

**Method.**    Similar to Section 3, to measure the change in training loss due to a particular learning algorithm, we partition the training set into $M$ subgroups, each with a different range of EL2N scores. For each subgroup, we then compute the average change in training loss after we apply the learning algorithm, relative to the resulting training loss after training the dense model with standard hyperparameters as specified in Appendix B.2, on examples in the subgroup. A negative value indicates that the learning algorithm improves the subgroup training loss relative to training the dense model with standard hyperparameters. We present the effect of pruning on subgroup training loss across a sequence of evenly spaced 5 sparsities in Figure 18. We present the effect of extended dense training and width downscaling on subgroup training loss across a sequence of 5 evenly spaced training time milestones and model downscaling factors in Figure 19 and Figure 20, respectively. Consistent with Section 3, we pick $M = 20$ because it is the largest value of $M$ that enables us to clearly present per-subgroup training loss change.

**Pruning results.**    Figure 18 shows that, at low sparsities, pruning's effect is to improve training loss on almost all example subgroups. For example, on CIFAR-10-ResNet20 benchmark with 0% random label noise, pruning to 80% to 26.21% weights remaining reduces the average training loss on example subgroups. At these sparsities, pruning's effect is to improve training. Pruning to higher sparsities, pruning's regularization effect dominates – pruning increases training loss across all example subgroups. For example, on CIFAR-10-ResNet20 benchmark with 8.59% to 0.92% weights remaining, pruning's effect is to increase the average training loss on almost all example subgroups.

**Extended dense training results.**    Figure 19 shows that the effect of extended dense training is to improve training loss on almost all example subgroups. This effect is similar to pruning to low sparsities (e.g., 80% to 26.21% weights remaining for CIFAR-10-ResNet20), as shown in Figure 18. However, unlike pruning to high sparsities (e.g, 8.59% to 0.92% weights remaining for CIFAR-10-ResNet20), we do not observe any regularization effect of extended dense training.

**Width downscaling results.**    Figure 20 shows that width downscaling has a regularization effect, as it increases training loss on almost all example subgroups. This effect is similar to pruning to high sparsities (e.g., 8.59% to 0.92% weights remaining for CIFAR-10-ResNet20), as shown in Figure 18. However, unlike pruning to low sparsities (e.g, 80% to 26.21% weights remaining for CIFAR-10-ResNet20), we do not observe any training loss improvement at any model width we test.

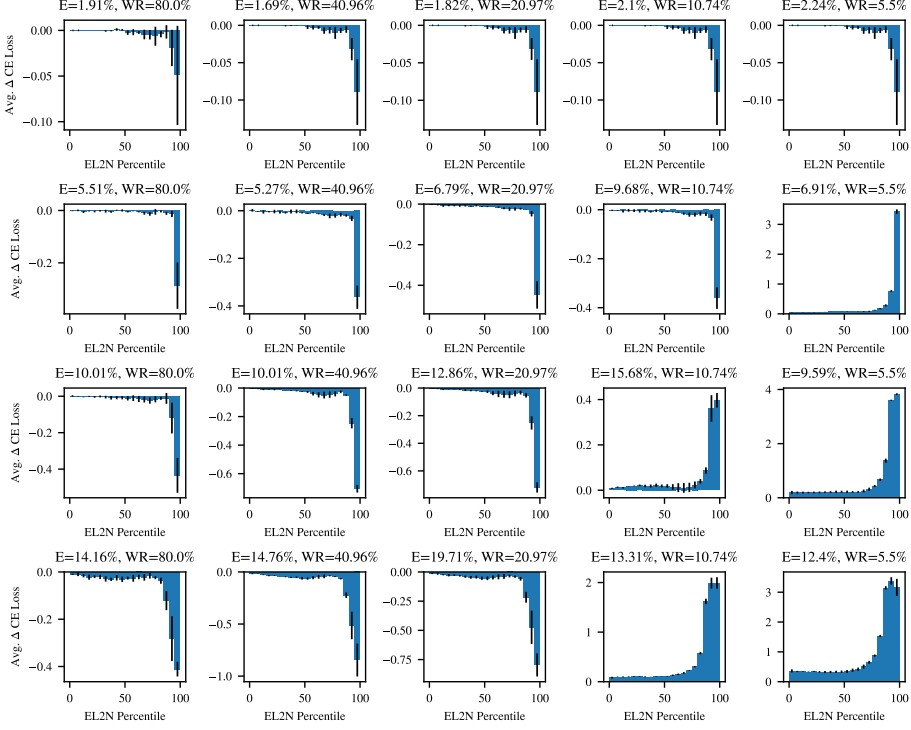

(a) LeNet-MNIST, rows correspond to 0%, 5%, 10% and 15% random label noise

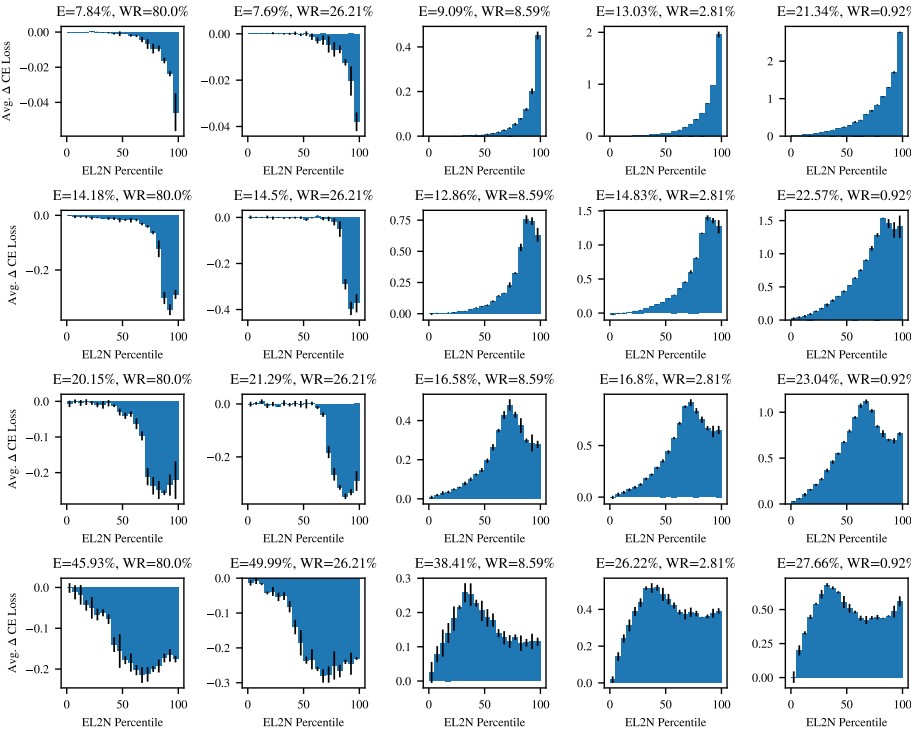

(b) CIFAR-10-ResNet20, rows correspond to 0%, 15%, 30% and 60% random label noise

Figure 18: Pruning's effect on average training loss on subgroups of examples with distinct EL2N score percentile range. Title shows the test error (E) and weights remaining (WR) of the sparse model. A negative value indicates that pruning improves training loss.

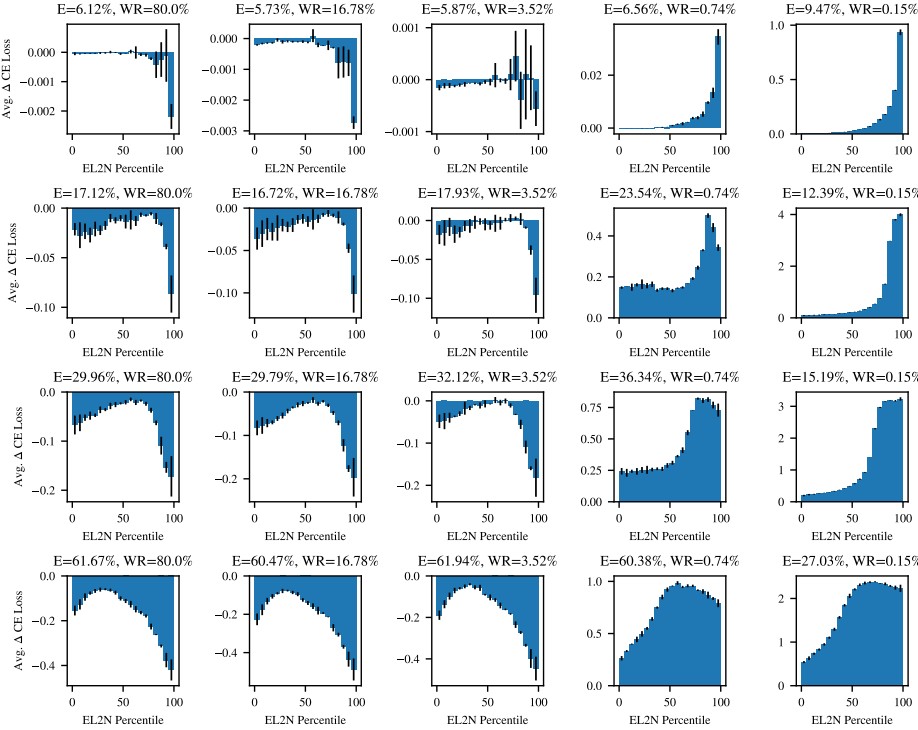

(c) CIFAR-10-VGG-16, rows correspond to 0%, 15%, 30% and 60% random label noise

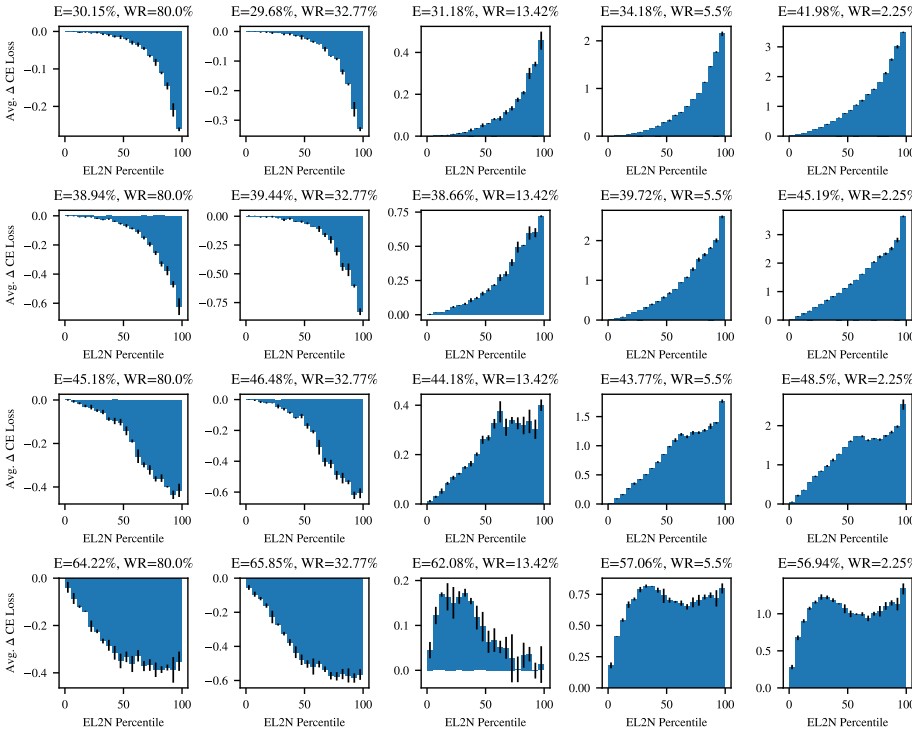

(d) CIFAR-100-ResNet32, rows correspond to 0%, 15%, 30% and 60% random label noise

Figure 18: (Cont.) Pruning's effect on average training loss on subgroups of examples with distinct EL2N score percentile range. Title shows the test error (E) and weights remaining (WR) of the sparse model. A negative value indicates that pruning improves training loss.

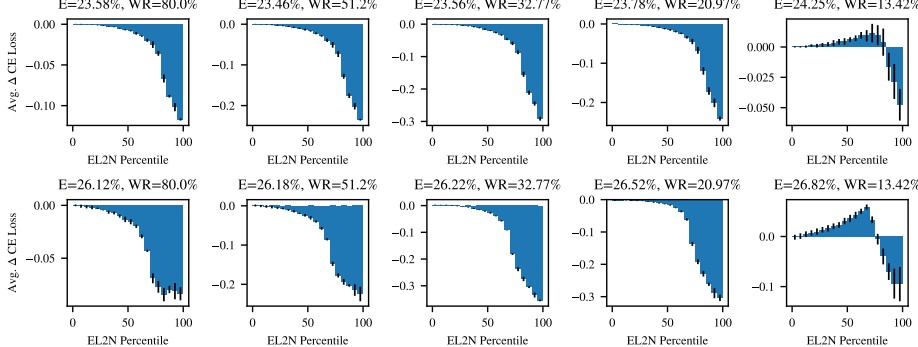

(e) ImageNet-ResNet50, with 0%, 15% random label noise

Figure 18: (Cont.) Pruning's effect on average training loss on subgroups of examples with distinct EL2N score percentile range. Title shows the test error (E) and weights remaining (WR) of the sparse model. A negative value indicates that pruning improves training loss.

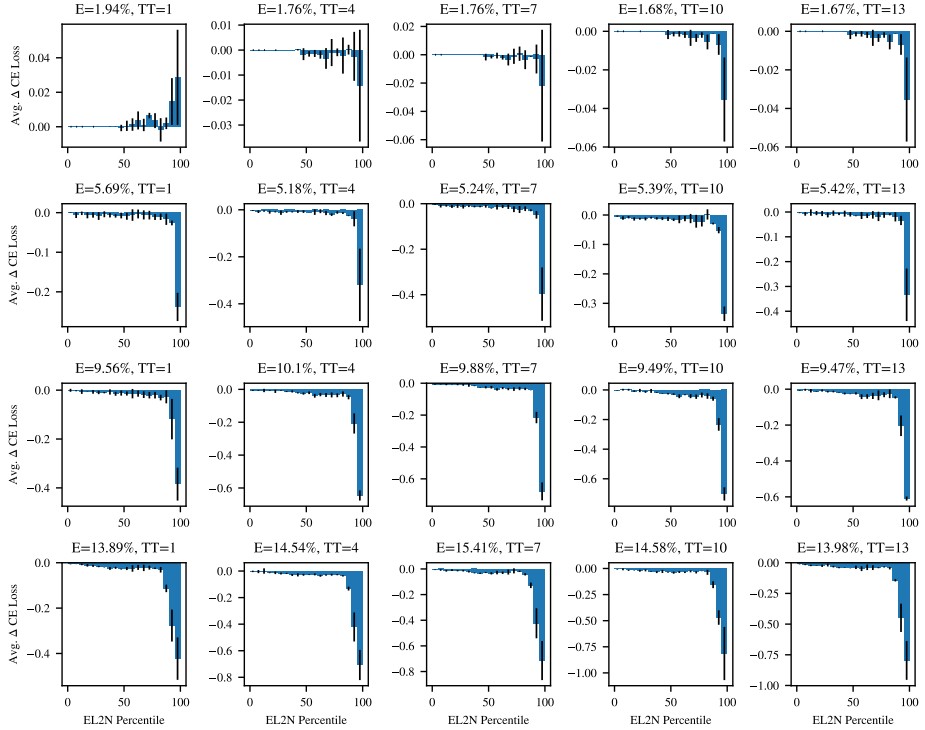

(a) LeNet-MNIST, rows correspond to 0%, 5%, 10% and 15% random label noise

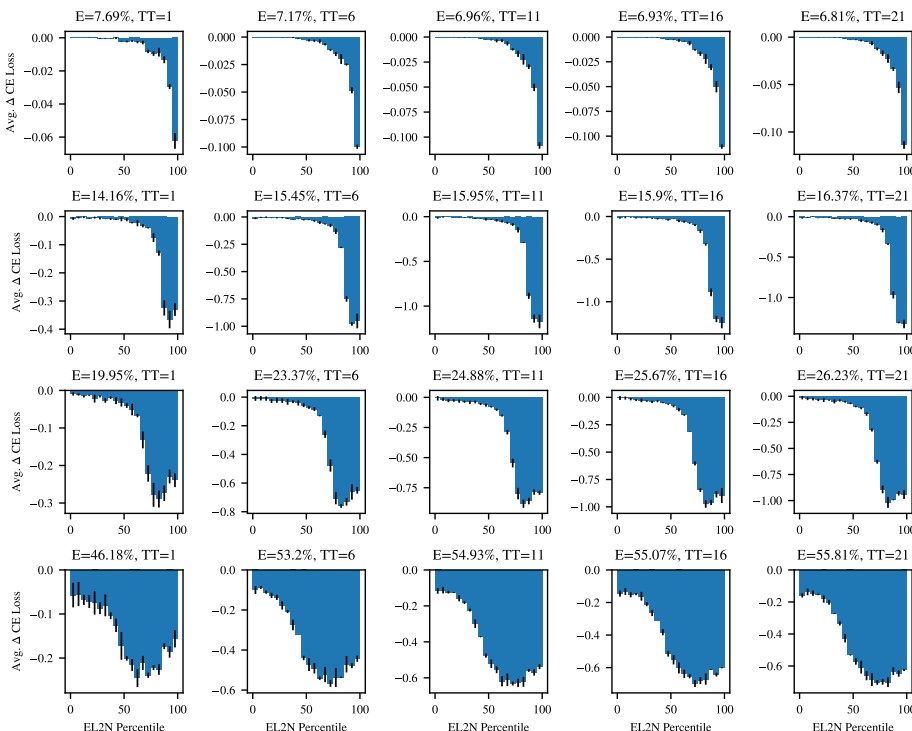

(b) CIFAR-10-ResNet20, rows correspond to 0%, 15%, 30% and 60% random label noise

Figure 19: Extended dense training improves subgroup training loss. Title shows the test error (E) and training time (TT) in number of pruning iterations' worth of training epochs. A negative value indicates that extended dense training improves training loss.

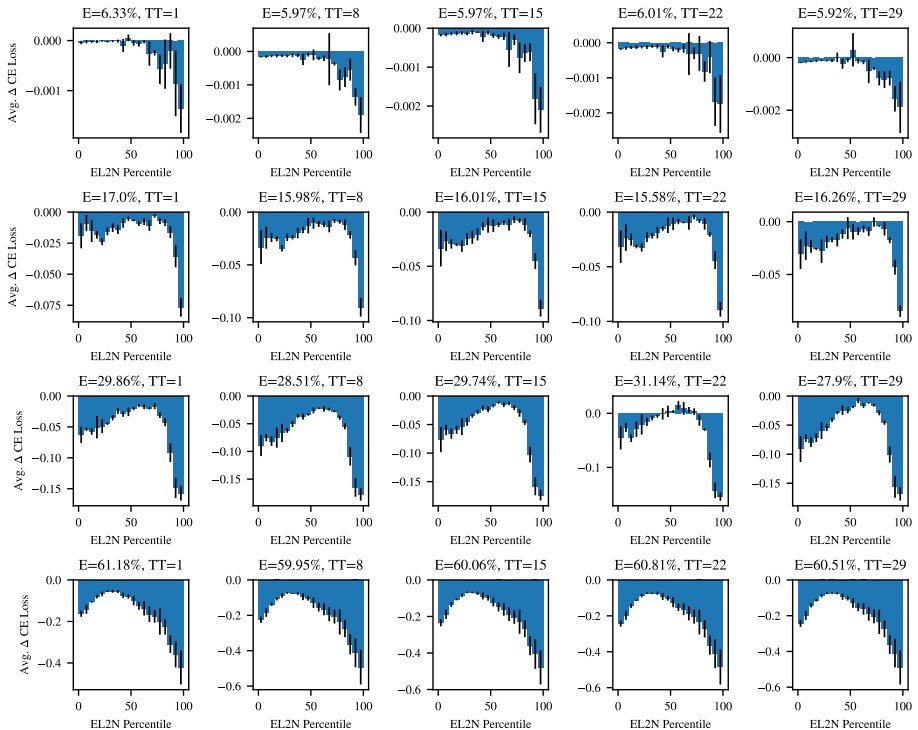

(c) CIFAR-10-VGG-16, rows correspond to 0%, 15%, 30% and 60% random label noise

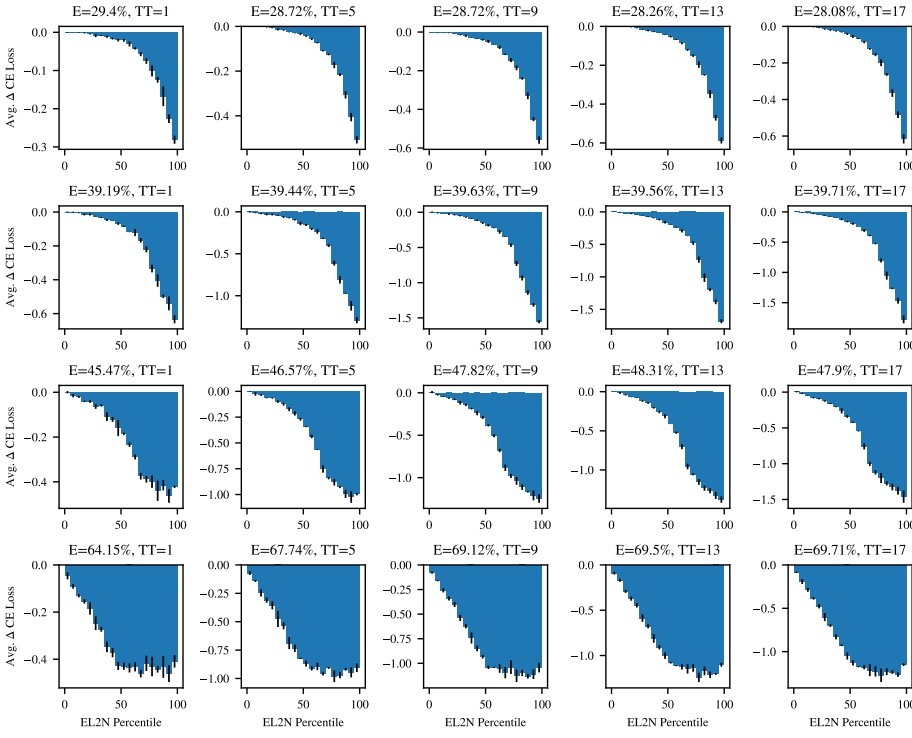

(d) CIFAR-100-ResNet32, rows correspond to 0%, 15%, 30% and 60% random label noise

Figure 19: (Cont.) Extended dense training improves subgroup training loss. Title shows the test error (E) and training time (TT) in number of pruning iterations' worth of training epochs. A negative value indicates that extended dense training improves training loss.

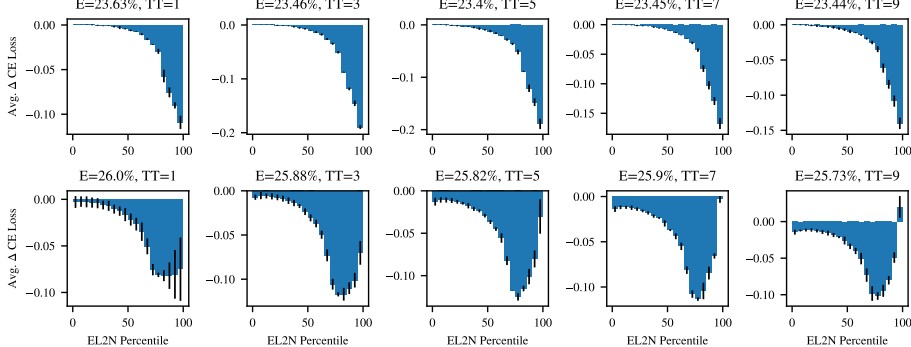

(e) ImageNet-ResNet50, with 0%, 15% random label noise

Figure 19: (Cont.) Extended dense training improves subgroup training loss. Title shows the test error (E) and training time (TT) in number of pruning iterations' worth of training epochs. A negative value indicates that extended dense training improves training loss.

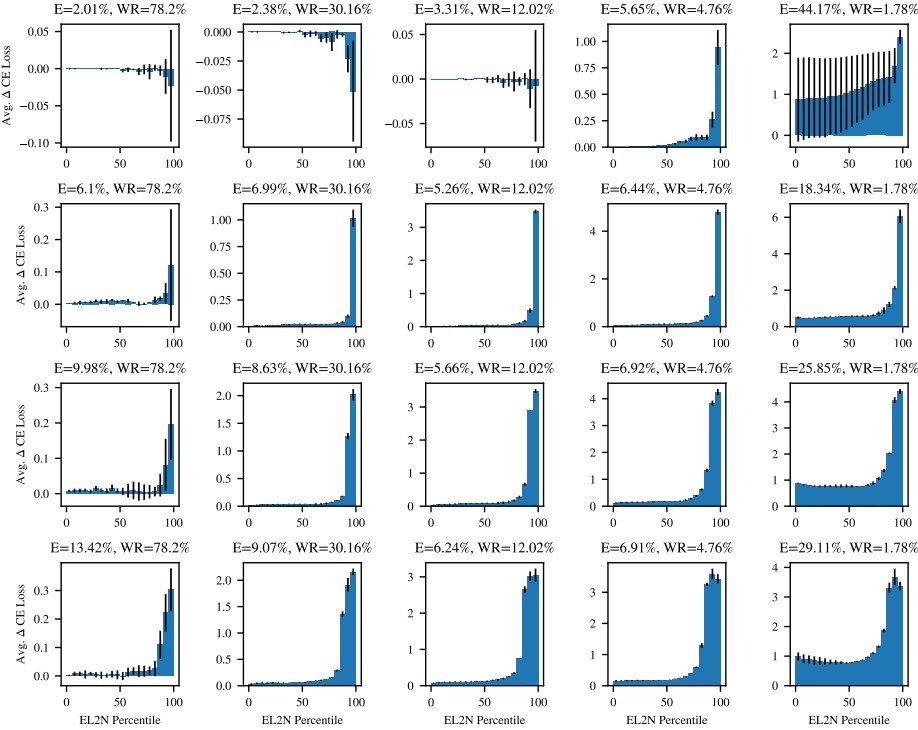

(a) LeNet-MNIST, rows correspond to 0%, 5%, 10% and 15% random label noise

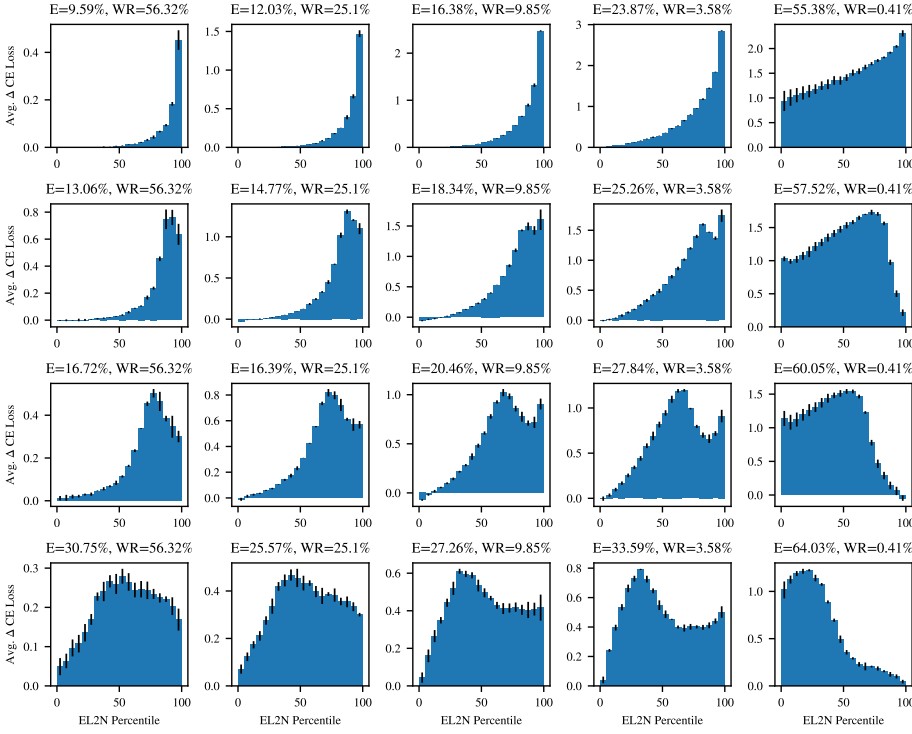

(b) CIFAR-10-ResNet20, rows correspond to 0%, 15%, 30% and 60% random label noise

Figure 20: Width downscaling increases subgroup training loss. Title shows the test error (E) and weights remaining (WR), as a portion of the number of parameters in the model with the original and unscaled width. A positive value indicates that width downscaling increases training loss.

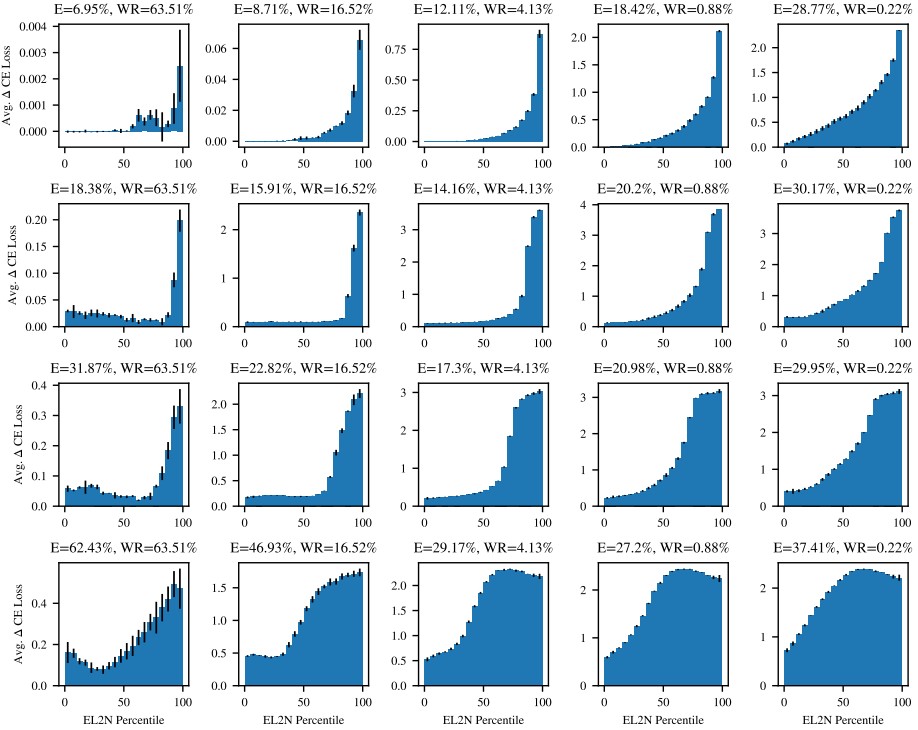

(c) CIFAR-10-VGG-16, rows correspond to 0%, 15%, 30% and 60% random label noise

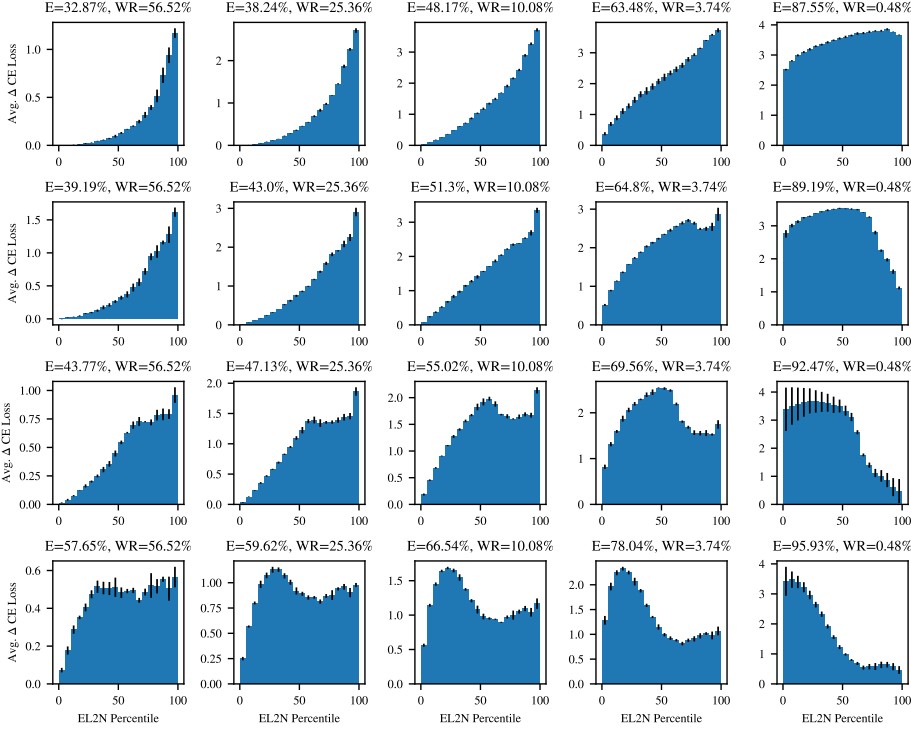

(d) CIFAR-100-ResNet32, rows correspond to 0%, 15%, 30% and 60% random label noise

Figure 20: (Cont.) Width downscaling increases subgroup training loss. Title shows the test error (E) and weights remaining (WR), as a portion of the number of weights in the model with the original and unscaled width. A positive value indicates that width downscaling increases training loss.

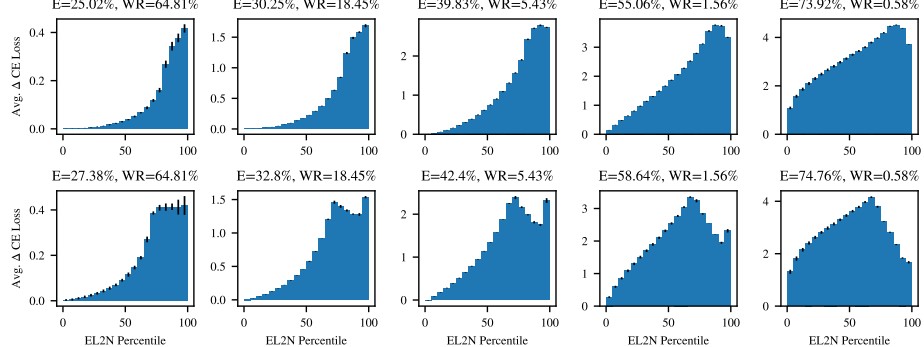

(e) ImageNet-ResNet50, with 0%, 15% random label noise

Figure 20: (Cont.) Width downscaling increases subgroup training loss. Title shows the test error (E) and weights remaining (WR), as a portion of the number of weights in the model with the original and unscaled width. A positive value indicates that width downscaling increases training loss.

| Models | Method | Noise Level | Test Errors |
|---|---|---|---|
| M-LeNet | LR | 5%/10%/15% | 5.21±0.08%/7.57±0.35%/9.96±0.62% |
| | EDT | 5%/10%/15% | 4.84±0.33%/8.96±0.13%/13.04±0.71% |
| C10-ResNet20 | LR | 15%/30%/60% | 13.04±0.43%/16.03±0.13%/25.39±0.11% |
| | EDT | 15%/30%/60% | 13.45±0.16%/18.40±0.24%/39.87±0.58% |
| C10-VGG-16 | LR | 15%/30%/60% | 12.47±0.29%/15.47±0.13%/25.29±0.49% |
| | EDT | 15%/30%/60% | 16.04±0.22%/28.22±0.37%/59.46±0.53% |
| C100-ResNet32 | LR | 15%/30%/60% | 38.81±0.70%/43.86±0.29%/55.83±0.45% |
| | EDT | 15%/30%/60% | 39.51±0.40%/44.10±0.49%/60.62±0.74% |
| I-ResNet50 | LR | 15% | 26.15±0.09% |
| | EDT | 15% | 25.95±0.06% |

Table 5: Comparing Learning Rate Rewinding with Extended Dense Training. LR=Learning Rate Rewinding, EDT=Extended Dense Training.

## M   Data for Experiments in Paper

| Model | Noise Level | Rewind | Type | Test Error |
|---|---|---|---|---|
| M-LeNet | 0%/5%/10%/15% | Weight | Sparse | 1.5±0.1%/3.7±0.3%/4.2±0.1%/4.4±0.2% |
| | | LR | Sparse | 1.8±0.1%/5.2±0.1%/7.6±0.3%/10.0±0.6% |
| | | N/A | Dense | 2.1±0.2%/5.3±0.2%/9.1±0.2%/13.4±0.3% |
| C10-ResNet20 | 0%/15%/30%/60% | Weight | Sparse | 8.1±0.3%/13.1±0.2%/15.4±0.2%/24.5±0.5% |
| | | LR | Sparse | 7.7±0.1%/13.0±0.4%/16.0±0.1%/25.4±0.1% |
| | | N/A | Dense | 8.2±0.1%/13.6±0.3%/18.4±0.4%/40.3±0.4% |
| C10-VGG-16 | 0%/15%/30%/60% | Weight | Sparse | 6.2±0.1%/12.0±0.3%/14.8±0.4%/23.8±0.2% |
| | | LR | Sparse | 5.9±0.1%/12.5±0.3%/15.5±0.1%/25.5±0.3% |
| | | N/A | Dense | 6.6±0.0%/17.5±0.2%/31.5±0.6%/62.7±0.6% |
| C100-ResNet32 | 0%/15%/30%/60% | Weight | Sparse | 30.6±0.4%/38.7±0.6%/42.9±0.1%/54.7±0.5% |
| | | LR | Sparse | 29.9±0.3%/38.8±0.7%/43.9±0.3%/55.8±0.5% |
| | | N/A | Dense | 30.3±0.2%/38.9±0.3%/44.0±0.6%/61.0±0.4% |

Table 6: Pruning's Impact on Model Generalization. Shorthand for dataset names: M=MNIST, C10=CIFAR-10, C100=CIFAR-100, I=ImageNet.

| Model | Method | Noise Level | Test Error |
|---|---|---|---|
| M-LeNet | WD | 0%/5%/10%/15% | 1.90±0.09%/5.63±0.18%/5.45±0.38%/6.08±0.33% |
|  | LR | 0%/5%/10%/15% | 1.82±0.11%/5.21±0.08%/7.57±0.35%/9.96±0.62% |
| C10-ResNet20 | WD | 0%/15%/30%/60% | 8.42±0.11%/13.06±0.16%/16.01±0.55%/25.29±0.59% |
|  | LR | 0%/15%/30%/60% | 7.71±0.09%/13.04±0.43%/16.03±0.13%/25.39±0.11% |
| C10-VGG-16 | WD | 0%/15%/30%/60% | 6.67±0.04%/14.32±0.24%/17.43±0.10%/26.76±0.37% |
|  | LR | 0%/15%/30%/60% | 5.86±0.08%/12.47±0.29%/15.47±0.13%/25.29±0.49% |
| C100-ResNet32 | WD | 0%/15%/30%/60% | 30.43±0.25%/38.88±0.13%/44.41±0.16%/57.92±0.29% |
|  | LR | 0%/15%/30%/60% | 29.93±0.34%/38.81±0.70%/43.86±0.29%/55.83±0.45% |
| I-ResNet50 | WD | 0%/15% | 23.89±0.10%/24.75±0.18% |
|  | LR | 0%/15% | 23.42±0.14%/26.15±0.09% |

Table 7: Comparing Pruning with Width Downscaling. Pruning matches or exceeds the generalization performance of width downscaling, except on MNIST-LeNet benchmark. Shorthand for dataset names: M=MNIST, C10=CIFAR-10, C100=CIFAR-100, I=ImageNet.

| Model | Method | Noise Level | Test Error (Optimal) | Test Error (0% Sparsity) |
|---|---|---|---|---|
| MNIST | LR | 5%/10%/15% | 5.22±0.06%/7.57±0.35%/10.03±0.60% | 5.35±0.18%/9.08±0.21%/13.40±0.32% |
| LeNet | DS | 5%/10%/15% | 2.55±0.12%/2.74±0.13%/3.30±0.11% | 5.23±0.35%/9.11±0.09%/12.03±0.93% |
| CIFAR-10 | LR | 15%/30%/60% | 12.63±0.12%/15.88±0.26%/25.39±0.11% | 13.57±0.26%/18.35±0.42%/40.26±0.43% |
| ResNet20 | DS | 15%/30%/60% | 9.76±0.11%/11.38±0.10%/18.66±0.26% | 11.20±0.29%/14.51±0.40%/32.77±0.48% |
| CIFAR-10 | LR | 15%/30%/60% | 12.39±0.27%/15.19±0.26%/25.49±0.29% | 17.51±0.18%/31.46±0.60%/62.68±0.63% |
| VGG-16 | DS | 15%/30%/60% | 8.84±0.08%/10.09±0.09%/17.80±0.37% | 18.07±0.57%/32.25±0.40%/61.21±0.88% |
| C100 | LR | 15%/30%/60% | 38.81±0.70%/43.29±0.40%/55.81±0.46% | 38.85±0.29%/43.97±0.61%/61.03±0.36% |
| ResNet32 | DS | 15%/30%/60% | 34.24±0.25%/37.17±0.23%/47.38±1.10% | 34.35±0.26%/37.89±0.33%/52.66±0.42% |
| ImageNet | LR | 15% | 26.12±0.08% | 26.44±0.06% |
| ResNet50 | DS | 15% | 24.75±0.09% | 24.77±0.07% |

Table 8: Comparing Pruning with Training Dense Models Exclusively on Dataset Subsets Predicted Correctly by Sparse Models (referred to as "Dense Subset" models). "Dense Subset" models matches or exceeds the generalization performance achieved by pruning. Shorthand for dataset names: LR = Learning Rate Rewinding, DS = "Dense Subset".

| Model | Method | Noise Level | Test Error |
|---|---|---|---|
| M-LeNet | WR | 0%/5%/10%/15% | 1.50±0.08%/3.74±0.30%/4.16±0.07%/4.41±0.16% |
|  | LR | 0%/5%/10%/15% | 1.82±0.11%/5.21±0.08%/7.57±0.35%/9.96±0.62% |
| C10-ResNet20 | WR | 0%/15%/30%/60% | 8.07±0.29%/13.09±0.24%/15.43±0.25%/24.47±0.49% |
|  | LR | 0%/15%/30%/60% | 7.71±0.09%/13.04±0.43%/16.03±0.13%/25.39±0.11% |
| C10-VGG-16 | WR | 0%/15%/30%/60% | 6.19±0.10%/11.97±0.33%/14.81±0.39%/23.80±0.15% |
|  | LR | 0%/15%/30%/60% | 5.86±0.08%/12.47±0.29%/15.47±0.13%/27.90±1.41% |
| C100-ResNet32 | WR | 0%/15%/30%/60% | 30.58±0.36%/38.74±0.59%/42.89±0.09%/54.67±0.50% |
|  | LR | 0%/15%/30%/60% | 29.93±0.34%/38.81±0.70%/43.86±0.29%/55.83±0.45% |
| I-ResNet50 | WR | 0%/15% | 23.51±0.08%/26.14±0.23% |
|  | LR | 0%/15% | 23.42±0.14%/26.15±0.09% |

Table 9: The Effect of Weight Resetting on Pruning's Generalization Improvement. Shorthand for dataset names: M=MNIST, C10=CIFAR-10, C100=CIFAR-100, I=ImageNet.

| Model | Method | Noise Level | Test Error |
|---|---|---|---|
| M-LeNet | Magnitude | 0%/5%/10%/15% | 1.82±0.11%/5.21±0.08%/7.57±0.35%/9.96±0.62% |
| | SynFlow | 0%/5%/10%/15% | 1.65±0.05%/5.20±0.15%/9.59±0.20%/11.72±0.64% |
| | Random | 0%/5%/10%/15% | 1.86±0.19%/5.11±0.27%/6.88±0.23%/7.50±0.07% |
| C10-ResNet20 | Magnitude | 0%/15%/30%/60% | 7.71±0.09%/13.04±0.43%/16.03±0.13%/25.39±0.11% |
| | SynFlow | 0%/15%/30%/60% | 7.83±0.28%/12.61±0.41%/15.27±0.02%/25.09±0.20% |
| | Random | 0%/15%/30%/60% | 8.14±0.29%/12.09±0.08%/14.53±0.09%/24.52±0.42% |
| C10-VGG-16 | Magnitude | 0%/15%/30%/60% | 5.86±0.08%/12.47±0.29%/15.47±0.13%/25.29±0.49% |
| | SynFlow | 0%/15%/30%/60% | 6.14±0.09%/12.35±0.18%/15.04±0.21%/30.67±10.03% |
| | Random | 0%/15%/30%/60% | 6.40±0.17%/14.65±0.39%/17.77±0.42%/27.30±0.49% |
| C100-ResNet32 | Magnitude | 0%/15%/30%/60% | 29.93±0.34%/38.81±0.70%/43.86±0.29%/55.83±0.45% |
| | SynFlow | 0%/15%/30%/60% | 30.24±0.24%/38.13±0.33%/43.12±0.57%/55.55±0.17% |
| | Random | 0%/15%/30%/60% | 30.15±0.14%/37.18±0.32%/41.44±0.30%/53.60±0.07% |
| I-ResNet50 | Magnitude | 0%/15% | 23.42±0.14%/26.15±0.09% |
| | SynFlow | 0%/15% | 23.90±0.08%/26.37±0.10% |
| | Random | 0%/15% | 23.86±0.10%/26.49±0.10% |

Table 10: The Effect of Weight Selection Heuristics on Pruning's Generalization Improvement.
Shorthand for dataset names: M=MNIST, C10=CIFAR-10, C100=CIFAR-100, I=ImageNet.

| Benchmark | Noise Level | Partition | Dense/Sparse (Diff) CE Loss |
|---|---|---|---|
| M-LeNet | 5% | Noisy | 0.46±0.06/0.2±0.13(-0.26) |
| | | Original | 0.02±0.0/0.01±0.0(-0.01) |
| | 10% | Noisy | 0.53±0.04/3.83±0.06(+3.3) |
| | | Original | 0.03±0.01/0.24±0.01(+0.21) |
| | 15% | Noisy | 0.59±0.0/3.5±0.17(+2.91) |
| | | Original | 0.05±0.01/0.36±0.04(+0.31) |
| C10-ResNet20 | 15% | Noisy | 2.63±0.06/3.18±0.27(+0.55) |
| | | Original | 0.18±0.0/0.26±0.03(+0.08) |
| | 30% | Noisy | 2.52±0.01/3.1±0.03(+0.58) |
| | | Original | 0.34±0.0/0.55±0.02(+0.21) |
| | 60% | Noisy | 2.21±0.01/2.61±0.0(+0.4) |
| | | Original | 0.77±0.01/1.16±0.0(+0.39) |
| C10-VGG-16 | 15% | Noisy | 0.06±0.01/3.28±0.39(+3.22) |
| | | Original | 0.03±0.01/0.31±0.02(+0.28) |
| | 30% | Noisy | 0.12±0.01/2.99±0.1(+2.87) |
| | | Original | 0.07±0.0/0.53±0.01(+0.46) |
| | 60% | Noisy | 0.27±0.01/2.55±0.01(+2.28) |
| | | Original | 0.14±0.01/1.17±0.04(+1.03) |
| C100-ResNet32 | 15% | Noisy | 4.19±0.03/5.71±0.04(+1.52) |
| | | Original | 0.43±0.0/0.81±0.01(+0.38) |
| | 30% | Noisy | 4.25±0.04/5.13±0.34(+0.88) |
| | | Original | 0.66±0.01/1.05±0.16(+0.39) |
| | 60% | Noisy | 4.02±0.02/4.94±0.04(+0.92) |
| | | Original | 1.24±0.02/2.13±0.03(+0.89) |
| I-ResNet50 | 15% | Noisy | 7.53±0.03/7.34±0.04(-0.19) |
| | | Original | 0.56±0.0/0.53±0.0(-0.03) |

Table 11: Pruning's Impact on Training Loss Incurred on Noisy/Original partitions of Dataset.
Shorthand for dataset names: M=MNIST, C10=CIFAR-10, C100=CIFAR-100, I=ImageNet.