# OpenReview forum: "Pruning’s Effect on Generalization Through the Lens of Training and Regularization"
_NeurIPS.cc/2022/Conference — NeurIPS 2022 Accept_

### Official Review · Reviewer_wS1n · 2022-07-03

**Rating:** 6
**Confidence:** 4
**Soundness:** 3 good
**Presentation:** 3 good
**Contribution:** 3 good

**Summary:**

This paper provides emperical study on the effect of pruning on the generalization of neural network model. The discussion focus on state of the art iterative pruning pipeline, and discuss the effect of pruning on achieving better optimization and better regualrization for the model training process.

**Questions:**

1. Does longer training mainly contributes to better optimization, and smaller model mainly for stronger regularization?
2. What would be the effect in Figure 7 be if the weight is pruned in a elementwise fashion with the same zero positions as the fully trained sparse model from iterative pruning?

**Limitations:**

The limitation and social impact is adequately discussed.

**Strengths And Weaknesses:**

Strengths:
1. Understanding the effect of pruning on model generalization is an interesting and important topic, this paper makes insightful contribution along the line.
2. The paper is overall well written and easy to follow.
3. The paper make novel contribution by subgrouping the data with the difficulty of learning, and make seperated observation for data of different EL2N percentile. This leads to new insight on pruned model's optimization outcome and can inspire future investigation on other neural network optimization techniques.

Weaknesses:
1. The paper introduces two effects: better optimization and stronger regularization, and two possible reasons: longer training and reduced model size, but doesn't make clear discussion on what's the relationship between the reasons and effects. It would be better to discuss if (following common belief) longer training mainly leads to better optimization, and smaller model size mainly leads to stronger regularization; or the effect of the two potential reasons cannot be distinguished
2. The experiment setting on model size reduction may be improved: Since the work is mainly based on the pruning method used in lottery ticket hypothesis I believe the discussion should be focusing on element-wise pruning, yet the model size reduction experiment is set for an channel-wise pruning scenario. A better experiment may be reinitialize the nonzero weight element of a sparse model and retrain the model for the same epochs, this will give a fairer comparison with previous iterative pruning observations

---

> ### Author Response · Authors · 2022-08-02
> **Response**
>
> We thank you for the valuable feedback! We are happy that you find our contributions insightful and our analysis novel! We address two of your concerns below.
>
> —------------------------------------------------------------------------------
> > The paper introduces two effects: better optimization and stronger regularization, and two possible reasons: longer training and reduced model size, but doesn't make clear discussion on what's the relationship between the reasons and effects. It would be better to discuss if (following common belief) longer training mainly leads to better optimization, and smaller model size mainly leads to stronger regularization; or the effect of the two potential reasons cannot be distinguished
>
> > Does longer training mainly contributes to better optimization, and smaller model mainly for stronger regularization?
>
> We agree and have edited our paper (specifically results paragraph in Sec 5.1 and 5.2) to reflect that longer training time mainly leads to better optimization and smaller model size mainly leads to stronger regularization. We provide the following evidence for such claims:
> In Appendix M, we plot the change in subgroup training CE loss before and after applying the following learning algorithms:
> - Extended training with a cyclic learning rate schedule (introduced in Sec 5.1).
> - Training with a model with smaller width (introduced in Sec 5.2).
> - Pruning (i.e., learning rate rewinding).
>
> We observe that:
> - Extended training leads to a non-uniform decrease in subgroup CE loss similar to pruning when its effect is to improve optimization.
> - Model size reduction through down-scaling model width leads to a non-uniform increase in subgroup CE loss similar to pruning when its effect is to strengthen regularization.
>
> > The experiment setting on model size reduction may be improved: Since the work is mainly based on the pruning method used in lottery ticket hypothesis I believe the discussion should be focusing on element-wise pruning, yet the model size reduction experiment is set for an channel-wise pruning scenario. A better experiment may be reinitialize the nonzero weight element of a sparse model and retrain the model for the same epochs, this will give a fairer comparison with previous iterative pruning observations
> > What would be the effect in Figure 7 be if the weight is pruned in a elementwise fashion with the same zero positions as the fully trained sparse model from iterative pruning?
>
> We agree that comparing pruning with size-reduction techniques that similarly remove weights in an element-wise fashion can strengthen our conclusion in Sec 5.2. For reference our conclusion in Sec 5.2 is:
>
> - Size reduction alone, through down-scaling model width, cannot explain changes in generalization due to pruning.
>
> To resolve your concern, we run additional experiments to compare pruning with training from scratch with an equivalent amount of sparsity induced by randomly removing weights at initialization (referred to as training with random sparsity below). Training with random sparsity reduces model size by pruning away weights in an element-wise fashion, similar to pruning.
>
> Our results show that our conclusion continues to hold even if we replace width scaling with training with random sparsity. We find that on standard datasets without random label noise, size-reduction via training with random sparsity either does not improve generalization at all or improves it by a smaller extent compared with pruning. On MNIST, training with random sparsity improves generalization by 0.14% (falling short of 0.3% improvement from iterative magnitude pruning), training with random sparsity does not improve generalization on CIFAR10-ResNet20, CIFAR10-VGG16, CIFAR100-ResNet32 benchmarks. Results for ImageNet-ResNet50 benchmarks are still running, as well as results for benchmarks with random label noise and we will report back ASAP. However, the existence of aforementioned benchmarks where size-reduction does not improve generalization to the same extent as pruning, even though both algorithms reduces model size by pruning weights elementwise, already demonstrate that size-reduction does not fully explain pruning’s benefits to generalization, suggesting that our conclusion in Sec 5.2 is valid.

---

> > ### Comment · Reviewer_wS1n · 2022-08-07
> > **Thanks for the response**
> >
> > I would like to thank the author for the responses and updates. These responses further strengthen the paper. Thus I will keep my score for acceptance.

---

> > > ### Author Response · Authors · 2022-08-09
> > > **Follow-up**
> > >
> > > > Thus I will keep my score for acceptance.
> > >
> > > Given that the threshold for acceptance is unknown, we would be eager to carry out any changes or address any concerns that are preventing you from being comfortable to recommend full (7) rather than weak (6) acceptance.

---

> > > ### Author Response · Authors · 2022-08-09
> > > **Results Follow-up**
> > >
> > > I hope to give you an update on the remaining experiments we promised to run comparing pruning with training with random sparsity.
> > >
> > > > The experiment setting on model size reduction may be improved: Since the work is mainly based on the pruning method used in lottery ticket hypothesis I believe the discussion should be focusing on element-wise pruning, yet the model size reduction experiment is set for an channel-wise pruning scenario. A better experiment may be reinitialize the nonzero weight element of a sparse model and retrain the model for the same epochs, this will give a fairer comparison with previous iterative pruning observations What would be the effect in Figure 7 be if the weight is pruned in a elementwise fashion with the same zero positions as the fully trained sparse model from iterative pruning?
> > >
> > > For reference, the conclusion we draw from the above experiment is the follow:
> > >
> > > > Size reduction alone, through down-scaling model width, cannot explain changes in generalization due to pruning.
> > >
> > > In Appendix P.3 we show that the conclusion in Sec 5.2 generalizes successfully to another model size-reduction technique - training with random sparsity. Notably, training with random sparsity reduces model size without reducing feature map size.
> > >
> > > **Method**. We compare pruning with training from scratch with an equivalent amount of random sparsity (which we refer to as **training with random sparsity** below).To induce random sparsity, we zero out a random set of weights at initialization. Training with random sparsity reduces model size in a element-wise fashion, enabling fair comparison between the effects of pruning and the effects of model size reduction on generalization.
> > > We also include comparison between pruning and width scaling as in Sec 5.2.
> > >
> > > **Results**. We plot and tabulate our results in Appendix P.3. We observe that neither training with random sparsity or width down-scaling emerges as the clear winner to achieve the most generalization improvement. For example, on CIFAR10-VGG16 with 60\% noise, width down-scaling **out-performs** training with random sparsity by 4.77\%, whereas on CIFAR100-ResNet32 with 60\% noise, width down-scaling **under-performs** training with random sparsity by 1.66\%. We therefore compare pruning with both width down-scaling and training with random sparsity.
> > >
> > > Out of 17 benchmarks completed, pruning out-performs both width downscaling and training with random sparsity on 9 benchmarks. Notably, on standard benchmarks without random label noise where pruning to optimal sparsity improves optimization, pruning always outperforms width down-scaling and training with random sparsity.
> > >
> > > **Conclusion**. Model size reduction alone, either through width down-scaling or through training with random sparsity, does not explain the full extent of the generalization-improving effect of pruning.

---

### Official Review · Reviewer_iZ5z · 2022-07-07

**Rating:** 6
**Confidence:** 3
**Soundness:** 3 good
**Presentation:** 4 excellent
**Contribution:** 3 good

**Summary:**

The paper finds that the size-reduction hypothesis can not fully explain the pruning's effects on generalization increase. It proposes the decomposition of such increase to better optimization with lower training loss and stronger regularization by avoiding hard training samples. Furthermore, the paper shows that neither extended training time nor size-reduction alone can not fully explain the generalization improvement. Thorough experiments are conducted to support the claim with the help of the EL2N score.

**Questions:**

I think it is important to show the training accuracy of width down-scaling methods in Figure 7. If the training accuracy of the two methods is far apart, the validity of the conclusion may be undermined. Since if there is a great training accuracy drop after using width down-scaling, there is no point talking about generalization.

Also see my concerns in weaknesses.

[Minor issues:]
1. It would be better to also include the optimal sparsity level for Figure 3.
2. Maybe it is better to give a further explanation of the augmented training method for Figure 1. If I understand correctly, % weights remaining is always 100% for the augmented training method. However, augmented training is also plotted using changing % weights remaining as the x-axis.

**Limitations:**

Yes, the authors have addressed the limitations and potential negative societal impact of their work.

**Strengths And Weaknesses:**

Strengths:
1. Most of the claims in the papers are backed by extensive experiments.
2. The paper is easy to follow.
3. It is a very interesting phenomenon to me that pruning actually improves optimization at a low sparsity ratio. Since most pruning papers mainly focus on test accuracy alone.
4. The decomposition of pruning's effect into better optimization and stronger regularization through the EL2N score is original and important.

Weaknesses:

1. For section 5.2, the comparison actually seems a bit unfair to me. First, the down-scaling width method, though having the same number of parameters compared to the weight pruning method, may have much less representation power (model capacity) due to lower feature map size. Second, the granularity of down-scaling width methods is not fine enough. For example, only 5 sparsity ratios are tested for the C100-ResNet32 combination while there are 10 for weight pruning methods. We may see a matching/better performance on datasets with random label noise.

2. For Figure 2, I think it is also helpful to plot average CE loss before and after pruning. It may give a better visualization if the neural networks avoid fitting difficult examples.

---

> ### Author Response · Authors · 2022-08-02
> **General Response**
>
> We thank you for the constructive feedback. We are happy to see that you recognize our efforts to corroborate our claims with extensive empirical evidence and that you appreciate the originality of our work!

---

> ### Author Response · Authors · 2022-08-02
> **Detailed Response**
>
> > For section 5.2, the comparison actually seems a bit unfair to me. First, the down-scaling width method, though having the same number of parameters compared to the weight pruning method, may have much less representation power (model capacity) due to lower feature map size.
>
> We agree that the reduced feature map size can be a confounding factor affecting our conclusion in Sec 5.2. For reference our conclusion in Sec 5.2 is:
> - Size reduction alone, through down-scaling model width, cannot explain changes in generalization due to pruning.
>
> To resolve your concern, we run additional experiments to compare pruning with training from scratch with an equivalent amount of sparsity induced by randomly removing weights at initialization (referred to as training with random sparsity below). Training with random sparsity reduces model size without changing feature map sizes.
>
> Our results show that our conclusion continues to hold even if we replace width scaling with training with random sparsity. We observe that on standard datasets without random label noise, size-reduction via training with random sparsity either does not improve generalization at all or improves it by a smaller extent compared with pruning. On MNIST, training with random sparsity improves generalization by 0.14% (falling short of 0.3% improvement from iterative magnitude pruning), training with random sparsity does not improve generalization on CIFAR10-ResNet20, CIFAR10-VGG16, CIFAR100-ResNet32 benchmarks. Results for ImageNet-ResNet50 benchmarks are still running, as well as results for benchmarks with random label noise and we will report back ASAP. However, the existence of aforementioned benchmarks where size-reduction does not improve generalization to the same extent as pruning, even when controlling for feature map size, already demonstrate that size-reduction does not fully explain pruning’s benefits to generalization, suggesting our conclusion in Sec 5.2 is valid.
>
> > Second, the granularity of down-scaling width methods is not fine enough. For example, only 5 sparsity ratios are tested for the C100-ResNet32 combination while there are 10 for weight pruning methods. We may see a matching/better performance on datasets with random label noise.
>
> We submitted additional experiments to test more model widths, and we will update our draft as soon as results become available.
>
> > For Figure 2, I think it is also helpful to plot average CE loss before and after pruning. It may give a better visualization if the neural networks avoid fitting difficult examples.
>
> We show the requested plot in the Appendix. P.1. We believe this alternative presentation enables us to arrive at similar conclusions as our original plots. If the reviewers feel strongly about this alternative presentation we are happy to switch over to them.
>
> > I think it is important to show the training accuracy of width down-scaling methods in Figure 7. If the training accuracy of the two methods is far apart, the validity of the conclusion may be undermined. Since if there is a great training accuracy drop after using width down-scaling, there is no point talking about generalization.
>
> We show the requested plot in Appendix P.2. Though we do not understand why the training accuracy gap between pruning and width scaling undermines the validity of our conclusion in Sec 5.2. For reference, here is our conclusion in Sec 5.2:
>
> - Size reduction alone, through down-scaling model width, cannot explain changes in generalization due to pruning.
>
> In almost all training error plots presented in Appendix P.2, we observe that pruning improves training loss at low sparsities whereas width down-scaling does not. This is consistent with our conclusion that size-reduction alone does not fully explain pruning’s generalization-improving effect – a key benefit of pruning is that it improves training loss at low sparsities; this benefit, however, is absent from size reduction through training with random sparsity.
>
> We also hope to clarify that the focus of our study is generalization error (as opposed to generalization gap, defined as the difference between train/test error). In the first paragraph of the paper, we specify that we measure generalization by test error.
>
> [Minor issues:]
> > It would be better to also include the optimal sparsity level for Figure 3.
>
> We have applied your suggestion.
>
> > Maybe it is better to give a further explanation of the augmented training method for Figure 1. If I understand correctly, % weights remaining is always 100% for the augmented training method. However, augmented training is also plotted using changing % weights remaining as the x-axis.
>
> Sorry for the confusion. We used two x-axis, one above marked with training time and one below marked with weights remaining. The one above is for augmented training, the one below for pruning. We insert an explanation for how to read the axis into the caption of Fig. 1.

---

> > ### Comment · Reviewer_iZ5z · 2022-08-03
> > **Response to the authors**
> >
> > 1. I thank the authors for the additional experiments. However, could the authors clarify where the results are?
> > 2. Thanks! I will wait for the results.
> > 3. Thanks for the results. I believe such visualization in the appendix may help the readers have a better understanding of the claims.
> > 4. I thank the authors for clarifying the difference between generalization gap and generalization error. It makes more sense now.

---

> > > ### Author Response · Authors · 2022-08-09
> > > **Follow-up**
> > >
> > > Apologies for the wait as we gather more results to show you, we address 1. and 2. below:
> > >
> > > > First, the down-scaling width method, though having the same number of parameters compared to the weight pruning method, may have much less representation power (model capacity) due to lower feature map size.
> > >
> > > > I thank the authors for the additional experiments. However, could the authors clarify where the results are?
> > >
> > > For reference, the conclusion we draw from the above experiment is the follow:
> > >
> > > > Size reduction alone, through down-scaling model width, cannot explain changes in generalization due to pruning.
> > >
> > > In Appendix P.3 we show that the conclusion in Sec 5.2 generalizes successfully to another model size-reduction technique - training with random sparsity. Notably, training with random sparsity reduces model size without reducing feature map size.
> > >
> > > **Method**. We compare pruning with training from scratch with an equivalent amount of random sparsity (which we refer to as **training with random sparsity** below).To induce random sparsity, we zero out a random set of weights at initialization. Training with random sparsity reduces model size without changing feature map sizes, enabling fair comparison between the effects of pruning and the effects of model size reduction on generalization.
> > >
> > > We also include comparison between pruning and width scaling as in Sec 5.2. Notably, as per your suggestion, we increased the number of model widths we enumerate to either match the number of sparsity levels we enumerate for pruning or reach the maximum number of integer model widths allowed.
> > >
> > > **Results**. We plot and tabulate our results in Appendix P.3. We observe that neither training with random sparsity or width down-scaling emerges as the clear winner to achieve the most generalization improvement. For example, on CIFAR10-VGG16 with 60\% noise, width down-scaling **out-performs** training with random sparsity by 4.77\%, whereas on CIFAR100-ResNet32 with 60\% noise, width down-scaling **under-performs** training with random sparsity by 1.66\%. We therefore compare pruning with both width down-scaling and training with random sparsity.
> > >
> > > Out of 17 benchmarks completed, pruning out-performs both width downscaling and training with random sparsity on 9 benchmarks. Notably, on standard benchmarks without random label noise where pruning to optimal sparsity improves optimization, pruning always outperforms width down-scaling and training with random sparsity.
> > >
> > > **Conclusion**. Model size reduction alone, either through width down-scaling or through training with random sparsity, does not explain the full extent of the generalization-improving effect of pruning.
> > >
> > > > Second, the granularity of down-scaling width methods is not fine enough. For example, only 5 sparsity ratios are tested for the C100-ResNet32 combination while there are 10 for weight pruning methods. We may see a matching/better performance on datasets with random label noise.
> > > > Thanks! I will wait for the results.
> > >
> > > We increased the number of model widths we enumerate to either match the number of sparsity levels we enumerate for pruning or reach the maximum number of integer model widths allowed. The results can be found in the plot and table in Appendix P.3.
> > >
> > > We find our conclusions unaffected by this change. Pruning out-performs width scaling on 11 out of a total of 18 benchmarks. Notably, on the original training dataset without random label noise, pruning unanimously out-performs width-scaling. These results lead to the same conclusion we reached in Sec 5.2:
> > >
> > > >  Size reduction alone, through down-scaling model width, cannot explain changes in generalization due to pruning.
> > >
> > > We highlight the following quantitative changes to our results:
> > >
> > > **Original observation.** On benchmarks with random label noise, pruning does better on LeNet, ResNet20, VGG16, ResNet32 and ResNet50 benchmarks by -4 to 0.4\%, 0\% (matching), 1.5 to 1.8\%, 0 (matching) to 2.1\%, and 0.4\% in test error, respectively.
> > >
> > > **New observation.** On benchmarks with random label noise, pruning does better on LeNet, ResNet20, VGG16, ResNet32 and ResNet50 benchmarks by -4 to 0\% (matching),  0\% (matching), 1.3 to 1.6\%, 0 (matching) to 2.6\%, and 0.3\% in test error, respectively.
> > >
> > > **Comment.** The gap between optimal generalization attained by pruning and width scaling shrinks noticeably on some benchmarks (e.g., MNIST-LeNet, VGG16) as we enumerate through additional model widths, however, the difference does not affect the conclusion of Sec 5.2 that model size reduction does not fully explain pruning’s generalization-improving effect.
> > > Note that the gap between pruning and width scaling grew on CIFAR100-ResNet32 even though we explored more model widths.
> > > This is because the optimal model is selected based on validation set performance.
> > > Trying more model widths leads to better validation set performance, which does not necessarily translate to a better test set performance (i.e., generalization).

---

### Official Review · Reviewer_TyWA · 2022-07-10

**Rating:** 7
**Confidence:** 5
**Soundness:** 3 good
**Presentation:** 3 good
**Contribution:** 3 good

**Summary:**

This paper investigates what makes network pruning improve accuracy despite the removal of parameters from the model. The authors contend that the folkloric claim about smaller models implying better generalization is not sufficient given the recent studies about the overparameterized regime, in which increasing model size may also improve generalization. As an alternative, the authors investigate the effect of pruning directly on the samples used for training by using a metric from the literature to identify how relevant each sample is for training the neural network. Consequently, they find out that pruning leads to a greater reduction in the loss function for samples that are more challenging to classify correctly, whereas pruning may also lead to an increase in the loss function for noisy samples that actually hurt generalization.

**Questions:**

0) Can you please give more consideration to the other modern lines of work on network pruning?

1) Your work heavily implies that Renda et al. (2020) is the state-of-the-art on network pruning. Can the authors cite other papers acknowledging that status? What about the extensive recent work on pruning at initialization, which spans way more than Frankle et al. (2021) (see Wang et al. (2020) + [7,10,13])?

2) The fact that a single algorithm is used for pruning is a noticeable limitation of the work. Have you tried other algorithms, even if something as simple as one-shot magnitude-based / gradient-based pruning, or - better yet - random pruning just to make a contrast?

3) I am not entirely convinced of the results in Section 5.2: the plot seems to indicate that there is a largely negative distinction for higher p when compared to dense models, and that the results are otherwise very similar. In line 269, how can you do better by -4 to 0.4%? Please explain.

4) The expression "better optimization" is very imprecise and in fact confusing (in line with people saying "more optimal"). The same about wording such as "the change in optimization due to pruning". Can the authors please consider a less ambiguous and more specific expression, such as "better training"?

5) Figure 1 (b): Have you measure these changes on the test set as well?

6) Your paper implies that negative changes due to the loss function are due to relevant but challenging examples, whereas positive changes are due to noisy examples. Whereas you have done some ablation of the noisy part with artificially inserted noise, have you looked into whether the samples for which the loss function change the most are indeed not the same as the pruned percentage goes higher and the changes become predominantly positive?

7) I am a bit curious about your thoughts on the rightmost plot of Figure 8 (a): Does that imply that MNIST is such an uniform dataset that the noisy discussion does not apply as much there?

8) In Section 4 and Figures 3-5, how did you choose the percentage p of random label noise? Why is it different across models/datasets? Can you add p=0% as another curve in each plot of Figures 3-5?

9) In the conclusion of Section 4, can you please describe in more detail how your computational results imply that low pruning levels are dominated by the optimization and that high pruning levels are more influenced by regularization?

10) What do you mean by pruning "stability" in line 297?

11) What "novel analytical technique" are you claiming in line 331?

12) How do you think your results relate to what happens in the case exact compression methods [14,15]?

**Limitations:**

The authors did a good job describing the limitations, except for using a single pruning algorithm to derive their conclusions.

**Strengths And Weaknesses:**

This paper presents a well organized and very compelling analysis on the effects of pruning on how the neural network learns from the training set. However, I was disappointed at how this paper, like many others in the LTH bandwagon, simply ignored the rest of the work currently being done in network pruning. Whereas the conclusions are very interesting, it makes me wonder what would happen if the same analysis was done with other pruning algorithms as well, or at least with random pruning for contrast.

I find the first two contributions very interesting, although I have questions below about how to distinguish more clearly between them. On the other hand, the third contribution is a bit of a stretch in my opinion: your work shows two factors that may partly explain what makes network pruning so useful. In that sense, they remain in equal footing with model reduction in terms of explaining something but not everything. I believe the wording is lines 92-94 is unnecessarily drastic. For example, the results in Section 5.2 make it clear to me that we actually cannot ignore the effect of model reduction in itself.

### Ignored work

The first paragraph cites modern papers that are exclusively based on magnitude-based pruning, but cites older work based on gradient-based pruning. The latter literature has grown into something that is often denoted as impact-based pruning and has remarkably good results as well. For example, the survey by Blalock et al. (2020) [1] outline these two kinds of approaches as predominant in the literature. With exception of  Wang et al. (2020), I see none of this second line of work acknowledged in the paper. I believe the authors should not ignore it, since it is a very active branch of the network pruning literature [2-14].

[1] https://arxiv.org/abs/2003.03033

[2] https://arxiv.org/abs/1506.02515

[3] https://arxiv.org/abs/1611.06440

[4] https://papers.nips.cc/paper/2017/hash/c5dc3e08849bec07e33ca353de62ea04-Abstract.html

[5] https://arxiv.org/abs/1711.05908

[6] https://openreview.net/forum?id=HJfwJ2A5KX

[7] https://openreview.net/forum?id=B1VZqjAcYX

[8] https://arxiv.org/abs/1905.05934

[9] https://arxiv.org/abs/2001.00218

[10] https://arxiv.org/abs/1906.06307

[11] https://arxiv.org/abs/1911.07412

[12] https://arxiv.org/abs/2004.14340

[13] https://arxiv.org/abs/2006.05467

[14] https://arxiv.org/abs/2102.07804

[15] https://arxiv.org/abs/2107.07467

[16] https://arxiv.org/abs/2203.04466

### Other comments about the writing of the paper

Abstract:
- I could not understand the last sentence of the abstract before reading the paper. I would recommend rewriting it, possibly breaking it down into multiple sentences.

Introduction:
- Line 38: Add space between word "generalization" and following citations
- Line 67: Remove first "but"
- Line 82: Remove "count reduction"

Section 2:
- Line 116: Remove second period

Section 3:
- Line 122: "are affected": add "more" between those words?
- Line 129: "he" -> "they" (that work has multiple authors)

Section 5:
- The title of this section is confusing, and I honestly was expecting something else when I read it: that you would try to isolate the effects of optimization and regularization from each other. Can you please change it to something more direct, such as "Isolating training time and model size effects"?

References:
- Han et al. (2015) is cited twice: remove one of those

Appendix B:
- Line 514: "interests" -> "interest"

---

> ### Author Response · Authors · 2022-08-02
> **General Response to TyWA**
>
> We thank you for the insightful feedback. We are excited that you find our analysis compelling and our conclusions interesting! We created Appendix O in the uploaded supplementary materials to host new results to show you.
>
> You expressed concern over our focus on a single magnitude based pruning algorithm. However, we analyzed 4 variants of pruning algorithms in Appendix B (old Appendix C) and provided results for both gradient based pruning algorithms as well as random pruning baseline. As per your detailed questions, we have now included further experimental results and analysis in the supplementary materials and provide you with a top-line summary as well as pointers below.
>
> Furthermore, we have taken actions to edit our paper introduction and related works to appropriately reflect the scope of pruning algorithms we consider in this paper and also cited modern, gradient-based/exact pruning algorithms to provide a more comprehensive account of the diverse set of modern pruning algorithms.

---

> > ### Comment · Reviewer_TyWA · 2022-08-07
> > **Following up**
> >
> > I would like to thank the reviewers for the careful follow up on my comments and for improving the paper in so many ways, especially with the additional experiments using other network pruning algorithms. This paper is a clear accept.
> >
> > With that said, let me just appeal once more for the authors to stop using the expression "better optimization", which can still be found 12 times in the paper: it hurts my eyes every time I read it. There is no such thing as "better optimization". I know that in at least one instance you have replaced the expression with a more suitable explanation about training, so all I am asking is that you find another shorter expression to refer to the same phenomenon elsewhere ("improved training", perhaps?). Please also take care of replacing "optimal optimization" with something more meaningful in Line 207. I do not think you want to alienate optimizers from seeing the value of your work.

---

> > > ### Author Response · Authors · 2022-08-09
> > > **Follow-up**
> > >
> > > We appreciate your feedback! We agree that "better/improved training" is strictly better than "better optimization" and applied your suggestion in our newly uploaded paper.

---

> ### Author Response · Authors · 2022-08-02
> **Detailed Response Part 1**
>
> > Whereas the conclusions are very interesting, it makes me wonder what would happen if the same analysis was done with other pruning algorithms as well, or at least with random pruning for contrast.
> > Can you please give more consideration to the other modern lines of work on network pruning?
> > The fact that a single algorithm is used for pruning is a noticeable limitation of the work. Have you tried other algorithms, even if something as simple as one-shot magnitude-based / gradient-based pruning, or - better yet - random pruning just to make a contrast?
>
> Our latest supplementary materials include results for SNIP and SynFlow (both of which are gradient based pruning algorithms). Our claims generalize successfully to these other pruning algorithms.
>
> **Methodology**.
> We apply the following pruning algorithms and random pruning for comparison:
> - For SNIP and SynFlow, we used their weight selection strategy to iteratively remove weights and retrain neural network models with surviving weights to compare fairly with learning rate rewinding.
> - Our definition of random pruning similarly entails iteratively removing model weights at random and retraining the model with surviving weights to compare fairly with learning rate rewinding.
> Below we re-iterate key claims we made in the paper and discuss them in the context of SNIP and SynFlow. The ImageNet-ResNet50 benchmark with SNIP pruning algorithm is still running so our discussion does not apply to this benchmark. ImageNet experiments can take weeks to complete so we will report them in follow-up comments.
>
> **Original claim**: Size-reduction does not fully explain pruning’s benefits to generalization.
>
> **Existing observation**: Recall that the extended dense training algorithm iteratively retrains the model without removing weights. On standard benchmarks without random label noise, we observe that the extended dense training algorithm improves generalization better than pruning with learning rate rewinding. We thus conclude that size-reduction does not fully explain pruning’s benefits to generalization.
>
> **New observations with alternative pruning algorithms**: Our claim generalizes successfully to other pruning algorithms. On standard benchmarks without random label noise, we observe that the extended dense training algorithm improves generalization better than SNIP, Synflow and random pruning algorithm. This observation implies that the size-reduction hypothesis is insufficient to fully explain pruning’s benefits to generalization, which is consistent with our observation with learning rate rewinding. For more details please refer to Appendix B.
>
> Our results also highlight that, on standard datasets, since pruning’s generalization-improving effect primarily results from optimization improvement, random pruning, when done iteratively, can also moderately improve generalization.
>
> **Original claim**: On standard datasets, pruning to optimal sparsity leads to better optimization.
>
> **Existing observation**: Pruning with learning rate rewinding reduces average training loss of all example subgroups with distinct EL2N score percentile ranges, particularly on example subgroups with high EL2N scores, suggesting better optimization.
>
> **New observations with alternative pruning algorithms**: Our claim generalizes successfully to other pruning algorithms. In Appendix O.1, we reproduced subgroup training loss change due to pruning using SNIP and Synflow at the sparsity achieving the best generalization on standard datasets. We find that at the aforementioned sparsity, both SNIP and Synflow improves generalization by improving training loss on all example subgroups with distinct EL2N score percentile ranges, with a particular emphasis on high EL2N examples, suggesting better optimization. This finding is consistent with our observation for learning rate rewinding (Sec. 3) that pruning to optimal sparsity leads to better optimization on standard datasets.
>
> We also show the same plot with random pruning for contrast. The difference is that on standard datasets and at optimal sparsity, random pruning sometimes improves generalization by increasing subgroup training loss, suggesting stronger regularization.

---

> > ### Author Response · Authors · 2022-08-02
> > **Rest of the Detailed Response Part 1**
> >
> > **Original claim:** On datasets with random label noise, pruning to optimal sparsity leads to stronger regularization.
> >
> > **Existing observation:** Pruning with learning rate rewinding increases average training loss of all example subgroups with distinct EL2N score percentile ranges, particularly on example subgroups with high EL2N scores.
> >
> > **New observations with alternative pruning algorithm:** Our claim generalizes successfully to other pruning algorithms. In Appendix O.2, we depict subgroup training loss change due to pruning using SNIP and SynFlow at the sparsity achieving the best generalization on datasets with random label noise. We find that at the aforementioned sparsity, SNIP and Synflow improves generalization by worsening training loss on all example subgroups with distinct EL2N score percentile ranges, suggesting stronger regularization. Both algorithms tend to affect high EL2N examples. This finding is consistent with our observation for learning rate rewinding (Sec. 4) that pruning to optimal sparsity leads to stronger regularization on datasets with random label noise.
> >
> > We show the same plot with random pruning and the results are qualitatively the same as SNIP and SynFlow.

---

> ### Author Response · Authors · 2022-08-02
> **Detailed Response Part 2**
>
> > the third contribution is a bit of a stretch in my opinion: your work shows two factors that may partly explain what makes network pruning so useful. In that sense, they remain in equal footing with model reduction in terms of explaining something but not everything.
> > I believe the wording is lines 92-94 is unnecessarily drastic.For example, the results in Section 5.2 make it clear to me that we actually cannot ignore the effect of model reduction in itself.
>
> We agree that model size reduction is indispensable to explaining pruning’s regularization effect and thus its benefits to generalization. In fact we followed up with Appendix I to discuss the similarities between pruning and width down-scaling. Based on this comparison, we believe that the regularization effect of pruning originates from size reduction.
>
> To address this confusion, we  propose to rephase our contribution.  For reference, here is, verbatim, our 3rd contribution from lines 92-94:
>
> - We demonstrate that our deconstruction of pruning’s effect into optimization and regularization cannot be further simplified — alternative explanations such as increased training time and reduced model size fail to fully explain the effects of pruning on generalization.
>
> We propose to rephase our contribution to the following, semantically equivalent statement:
>
> - We demonstrate that our deconstruction of pruning’s effect into optimization and regularization cannot be further simplified — alternative explanations such as increased training time and reduced model size **only partially explain** pruning’s generalization-improving effect.
>
>
> To clarify, better optimization and stronger regularization – to which model reduction contributes – fully account for pruning’s benefits to generalization. We provide the following evidence: measuring generalization improvement in terms of (average) test errors, we fully replicate pruning’s generalization improvement at optimal sparsity on dense models by improving optimization with extended training (Sec 5.1) and strengthening regularization with excluding examples that pruned models misclassify (Appendix D). There is no performance gap that remains to close.
>
>
>
> > Your work heavily implies that Renda et al. (2020) is the state-of-the-art on network pruning. Can the authors cite other papers acknowledging that status?
>
> In [1] from ICML 2021 the author states: [Iterative magnitude pruning] is a standard way to prune (Han et al., 2015) that gets state-of-the-art tradeoffs between error and unstructured density (Gale et al., 2019; Renda et al., 2020).
>
> > What about the extensive recent work on pruning at initialization, which spans way more than Frankle et al. (2021) (see Wang et al. (2020) + [7,10,13])?
>
> On standard datasets (without injecting random label noise), at optimal sparsity, pruning with learning rate rewinding achieves matching or better generalization improvement compared with pruning using SNIP and Synflow. See Table.2 in Appendix B for numeral results.

---

> ### Author Response · Authors · 2022-08-02
> **Detailed Response Part 3**
>
> > I am not entirely convinced of the results in Section 5.2: the plot seems to indicate that there is a largely negative distinction for higher p when compared to dense models, and that the results are otherwise very similar.In line 269, how can you do better by -4 to 0.4%? Please explain.
>
> We acknowledge that the presentation of Fig.7 is perhaps too cluttered for our analysis. We recall that our takeaway from Sec 5.2 is the following:
> - On standard benchmarks without label noise (p=0), pruning behaves differently than width scaling.
> - On benchmarks with random label noise (high p), pruning behaves similar to width scaling.
>
> Our takeaways are different because we measure similarity between algorithms differently. We focus on comparing the range of generalization improvements achieved by width scaling and pruning. Notably, we do not consider the parameter count at which particular generalization improvement occurs (i.e., parameter efficiency), as it falls outside the scope of our work.
>
> We show generalization improvement in absolute percentage at optimal sparsity and optimal model width for both algorithms below. Take C10-ResNet20 as an example, with no label noise (p=0%), pruning improves generalization by up to 0.84% while width scaling does not  improve generalization. We thus observe their difference on datasets with no label noise. At high label noise (e.g., p=60%), pruning and width scaling improves generalization by up to 14.87% and 14.69%. We thus conclude their similarity on datasets with random label noise.
>
> As for the negative improvements – on MNIST benchmarks, width scaling can actually out-perform pruning. We suspect this is because MNIST model training does not entail any explicit regularization, making it susceptible to overfitting caused by the optimization-improving effect of pruning.
>
> | Model         | Method | Noise Level    | Absolute Test Error Improvement (%) |
> |---------------|--------|----------------|-------------------------------------|
> | M-LeNet       | Prune  | 0%/5%/10%/15%  | 0.45/0.13/1.51/3.37                 |
> | M-LeNet       | WD     | 0%/5%/10%/15%  | 0.00/0.56/3.62/7.67                 |
> | C10-ResNet20  | Prune  | 0%/15%/30%/60% | 0.84/0.94/2.48/14.87                |
> | C10-ResNet20  | WD     | 0%/15%/30%/60% | 0.00/0.57/2.45/15.43                |
> | C10-VGG16     | Prune  | 0%/15%/30%/60% | 0.92/5.21/16.27/37.19               |
> | C10-VGG16     | WD     | 0%/15%/30%/60% | 0.00/3.60/14.53/36.14               |
> | C100-ResNet32 | Prune  | 0%/15%/30%/60% | 0.76/0.19/0.79/5.22                 |
> | C100-ResNet32 | WD     | 0%/15%/30%/60% | 0.00/0.00/0.53/2.50                 |
> | I-ResNet50    | Prune  | 0%/15%         | 0.49/0.38                           |
> | I-ResNet50    | WD     | 0%/15%         | 0.00/0.00                           |

---

> ### Author Response · Authors · 2022-08-02
> **Detailed Response Part 4**
>
> > The expression "better optimization" is very imprecise and in fact confusing (in line with people saying "more optimal"). The same about wording such as "the change in optimization due to pruning". Can the authors please consider a less ambiguous and more specific expression, such as "better training"?
>
> We agree that the word “optimization” is a loaded term and have tried our best to clarify phrases such as “the change in optimization due to pruning” as “the change in cross entropy loss on training examples due to pruning”. If you still feel strongly about replacing it with the phrase “better training”, we are happy to do so.
>
> > Figure 1 (b): Have you measure these changes on the test set as well?
>
> We show the requested plots in Appendix section O.3. Our observation that pruning to optimal sparsity affects CE loss of high EL2N (test) examples the most continues to characterize pruning’s effect on the test set. However the direction of influence is not quite predictable.: average CE loss is less meaningful on the test set because it is not the objective function being minimized by SGD (in contrast, average loss on the training set is the optimization objective). Unlike the training set, model weights are not optimized with respect to outliers in the test set and they may end up having very high CE loss since CE loss is unbounded. Averaging an array of unbounded quantities may thus give misleading results. We thus provide an extra plot showing changes in subgroup average test error due to pruning. Unlike CE loss, test error is bounded. We observe that pruning to optimal sparsity consistently improves test errors of high EL2N examples the most.
>
> > Your paper implies that negative changes due to the loss function are due to relevant but challenging examples, whereas positive changes are due to noisy examples. Whereas you have done some ablation of the noisy part with artificially inserted noise, have you looked into whether the samples for which the loss function change the most are indeed not the same as the pruned percentage goes higher and the changes become predominantly positive?
>
> Though we do not fully understand the question, we try our best to interpret it and give our answer. If our interpretation deviates from your intention, we appreciate your follow-up. We interpret your question as follows - say pruning to low sparsity L% improves average training loss of example subgroup A the most, and pruning to high sparsity H% worsens average training loss of example subgroup B the most.
> - Is subgroup A always challenging but relevant examples and B noisy examples? We do not make such an implication. In fact our results show otherwise - in Fig. 4, pruning to low sparsities improves training loss of noisy examples, too and can therefore worsen generalization while improving training loss. We do suggest that if A is predominantly challenging and relevant examples, pruning to L% should improve generalization.
> - Is subgroup A different from subgroup B? Our analysis does not find a notable difference between two subgroups. We visualized examples whose training loss are most improved/worsened by pruning in Appendix L, and we do not observe a distinction between examples experiencing the most significant training loss increase/decrease – they are both challenging to classify, and often contain noisy labels.
>
> Instead, the unifying observation we make is that pruning most significantly affects training loss of examples with high EL2N scores at all sparsities. Whether these high EL2N examples are challenging but relevant or noisy and detrimental to generalization determines the optimal sparsity, and thus the optimal combination of optimization/regularization effects to achieve the best generalization.
>
> > I am a bit curious about your thoughts on the rightmost plot of Figure 8 (a): Does that imply that MNIST is such an uniform dataset that the noisy discussion does not apply as much there?
>
> We believe this is a case where a pruned model suffers from connectivity issues, known as layer collapse [2]. The pruned model achieves a test error of 88.7%, which is very close to making random guesses. The training loss increase due to layer collapse occurs indiscriminately to every example thus appearing uniform.

---

> ### Author Response · Authors · 2022-08-02
> **Detailed Response Part 5**
>
> > In Section 4 and Figures 3-5, how did you choose the percentage p of random label noise? Why is it different across models/datasets?
>
> We use 15%, 30% and 60% label noise because we find that they generate visually distinct generalization curves. We switch to 5%, 10% and 15% on MNIST because generalization of LeNet tends to suffer easily in the presence of random label noise. This is likely due to the complete lack of explicit regularization during training of the LeNet model. On ImageNet we only used 15% to reduce compute cost as pruning ImageNet models with IMP takes weeks on SOTA non-distributed compute platforms.
>
> > Can you add p=0% as another curve in each plot of Figures 3-5?
>
> We’ve applied your suggestion.
>
> > In the conclusion of Section 4, can you please describe in more detail how your computational results imply that low pruning levels are dominated by the optimization and that high pruning levels are more influenced by regularization?
>
> Take C10-ResNet20 as an example, in Fig. 4, when pruning to low sparsities (100%-10% weights remaining), training loss on the noisy partition of the dataset actually improves by a small but noticeable extent. In the C10-ResNet20 plot in Fig. 4, this is shown as the dashed lines going down by a small amount between x=100% and x=10%. The decreased training loss suggests that pruning improves optimization.
>
> However, as we prune to more extreme sparsities (<10% weights remaining), Fig. 4 shows an uptick of training loss on both noisy and clean partitions of the dataset. In the C10-ResNet20 plot in Fig.4, this is shown as both dashed and solid lines going up as we go from x=10% to even smaller values of x. Furthermore, training loss tends to increase more on the noisy partition of the dataset than on the clean partition of the dataset. The increased training loss, which tends to affect examples with high EL2N scores the most, suggests stronger regularization.
>
> > What do you mean by pruning "stability" in line 297?
>
> Author of [3] defines it as “the drop in test accuracy immediately following [weights removal]”. We have revised the paper to clarify the definition of “stability”.
>
> > What "novel analytical technique" are you claiming in line 331?
>
> We refer to the technique of analyzing pruning’s effect on training dataset subgroups with distinct range of EL2N score percentiles to understand how pruning affects generalization.
>
> > How do you think your results relate to what happens in the case exact compression methods [14,15]?
>
> In the absence of experimental results, my hypothesis is that our observations do not apply to exact compression methods because exact compression methods produce a compressed network with the exact same behavior as the original.
>
> Both improved optimization and strengthened regularization are consequences of aspects of our pruning methods that produce a model with a different behavior from the original. The pruning algorithms we examine trains the pruned model with a cyclic learning rate, which improves optimization; retraining the pruned model with reduced size strengthens regularization. Since exact compression produces models with the same behavior as the original, our observed effects should not emerge.

---

> ### Author Response · Authors · 2022-08-02
> **References**
>
> [1] On the Predictability of Pruning Across Scales, ICML 2021.
>
> [2] Pruning neural networks without any data by iteratively conserving synaptic flow, NeurIPS 2020.
>
> [3] The Generalization-Stability Tradeoff In Neural Network Pruning, NeurIPS 2020.

---

### Meta-Review · Area_Chair_cs15 · 2022-09-04

**Recommendation:** Accept
**Confidence:** Certain

**Metareview:**

All the reviewers, including the AC, agree that the paper makes a significant contribution to the field that deserves publication at NeurIPS. The AC refers the readers to the reviews for the discussions on the pros and cons of the paper.

**Award:**

No

---

### Decision · Program_Chairs · 2022-09-14

Accept